# SeedGNN: Graph Neural Network for Supervised Seeded Graph Matching

## Abstract

There have been significant interests in designing Graph Neural Networks (GNNs) for seeded graph matching, which aims to match two (unlabeled) graphs using only topological information and a small set of seeds. However, most previous GNNs for seeded graph matching employ a semi-supervised approach, which requires a large number of seeds and can not learn knowledge transferable to unseen graphs. In contrast, this paper proposes a new supervised approach that can learn from a training set how to match unseen graphs with only a few seeds. At the core of our SeedGNN architecture are two novel modules: 1) a convolution module that can easily learn the capability of counting and using witnesses of different hops; 2) a percolation module that can use easily-matched pairs as new seeds to percolate and match other nodes. We evaluate SeedGNN on both synthetic and real graphs, and demonstrate significant performance improvement over both non-learning and learning algorithms in the existing literature. Further, our experiments confirm that the knowledge learned by SeedGNN from training graphs can be generalized to test graphs with different sizes and categories.

## 1 Introduction

Graph matching, also known as network alignment, aims to find the node correspondence between two graphs that maximally aligns their edge sets. As a ubiquitous but challenging problem, graph matching has numerous applications, including social network analysis (Narayanan et al., 2008; 2009; Zafarani et al., 2015; Zhang et al., 2015b;a; Chiasserini et al., 2016), computer vision (Conte et al., 2004; Schellewald et al., 2005; Vento et al., 2013), natural language processing (Haghighi et al., 2005), and computational biology (Singh et al., 2008; Kazemi et al., 2016; Kriege et al., 2019). This paper focuses on seeded graph matching, where a small portion of the node correspondence between the two graphs is revealed as seeds, and we seek to complete the correspondence by growing from the few seeded node pairs. Seeded graph matching is motivated by the fact that, in many real applications, the correspondence between a small portion of the two node sets is naturally available. For example, in social network de-anonymization, some users who explicitly link their accounts across different social networks could become seeds (Narayanan et al., 2008; 2009). Knowledge of even a few seeds has been shown to significantly improve the matching results for many real-world graphs (Kazemi et al., 2015; Fishkind et al., 2019).

Recently, the Graph Neural Network (GNN) approach for graph matching has attracted much research attention. Although such a machine-learning-based approach usually does not possess provable theoretical guarantees, it has the potential to learn valuable features from a large set of training data. Unfortunately, to date GNN has not been successfully applied to seeded graph matching. Most previous GNNs for seeded graph matching are limited to a *semi-supervised* learning paradigm, which only operates on a *single* pair of graphs (Zhang et al., 2019; Li et al., 2019a;b;c; Zhou et al., 2019; Chen et al., 2020; Derr et al., 2021) and treats the seed set as the labelled training data. The goal is to learn the useful features from the seed set, and then to generalize the knowledge to the rest of the unseeded nodes. This semi-supervised learning, however, suffers from two major limitations. First, in order to obtain high matching accuracy, the set of seeds needs to be sufficiently large, which is often unrealistic in practice. Second, as this semi-supervised setting only learns *within* a given pair of graphs, there is no effort in *transferring knowledge* from one pair of graphs to other pairs of unseen graphs, which severely limits GNNs' potential in distilling the common knowledge from a large set of training graphs. A natural but fundamental question is

*Can we learn to match two graphs from only a few seeds while generalizing to unseen graphs?*

This paper provides an affirmative answer to this question. Specifically, we design a novel GNN architecture through a supervised approach, namely SeedGNN, that can learn from many examples of matched graph pairs, distill the knowledge into the trained model automatically, and then apply such knowledge to match unseen graph pairs with only a few seeds. In contrast to prior GNN approach for seeded graph matching that apply GNNs *separately* to each graph and learn a *node-embedding* for each node (by aggregating neighborhood information within each individual graph), a key departure of our SeedGNN architecture is to apply the GNN *jointly* over two graphs and to learn a *pair-wise similarity* for each pair of nodes directly. As we will discuss further below, this *pair-wise* GNN architecture is crucial for learning both useful features (from seeds) and the best way to synthesize them in different types of graphs. (We note that this type of pair-wise GNNs have been used in a supervised learning approach for seedless graph matching in Rolínek et al. (2020); Wang et al. (2021). However, they have not been used for seeded graph matching. See Section 2 for further discussions.) Numerical experiments on both synthetic and real-world graphs show that our SeedGNN significantly outperforms the state-of-the-art algorithms, including both non-learning and learning-based ones, in terms of seed size requirement and matching accuracy. Moreover, our SeedGNN can generalize to match unseen graphs of sizes and types different from the training set.

At the core of our SeedGNN are two innovative designs. One is the convolution module that learns to count "witnesses" at different hops — a notion that plays a pivotal role in seeded graph matching (Mossel et al., 2019). Here, the $\ell$-hop witnesses of a node-pair are seeded pairs that lie in the neighborhood at $\ell$ hops. Naturally, a true pair is expected to have more witnesses than a fake pair. As we will further discuss in Section 4.1 and Section 4.2, our pair-wise SeedGNN architecture is much more effective than existing node-based GNNs in learning how to count witnesses, in a manner that can be easily generalized to unseen graphs. The second innovation is the percolation module that matches high-confidence node-pairs at one layer and propagates the matched node-pairs as new seeds to the subsequent layers, triggering a percolation process that matches a large number of node pairs. However, we emphasize that it remains highly non-trivial how to best utilize either the witness or the percolation idea for achieving high matching accuracy. Indeed, when graphs are very sparse, even true node-pairs may not have enough witnesses if the number of hops $\ell$ is small; when graphs are very dense, a fake pair may also have many witnesses if $\ell$ is large. Similarly, a fake pair may be incorrectly propagated as a new seed, which can lead to many cascading errors. The pair-wise architecture of SeedGNN is also crucial to facilitate learning how to best synthesize these two modules. As a result, our SeedGNN can potentially figure out which hops of witnesses are more reliable and what "cleaner" new seeds should be used to trigger the percolation process.

## 2 FURTHER RELATED WORK

**Theoretical Algorithms** Various seeded matching algorithms have been proposed based on hand-designed similarity metrics computed from local topological structures (Pedarsani et al., 2011; Yartseva et al., 2013; Korula et al., 2014; Chiasserini et al., 2016; Shirani et al., 2017; Mossel et al., 2019; Yu et al., 2021b). The theoretical analysis on these algorithms explains why a particular set of features (e.g., witnesses (Korula et al., 2014) and percolation (Yartseva et al., 2013)) are valuable for graph matching. However, these theoretical algorithms may not synthesize different features most effectively. See detailed discussion in Appendix A. In contrast, our SeedGNN can potentially figure out what combinations of features are most useful, and therefore it can potentially outperform known theoretical algorithms (see our experiments in Section 5 and Appendix C.3).

**GNN for Seedless Graph Matching** As we discussed earlier, most existing GNNs for seeded graph matching take a semi-supervised learning approach. In contrast, our SeedGNN falls into a supervised learning approach, which aims to transfer knowledge from training graphs to unseen graphs. In the literature, such a supervised learning approach has been applied to *seedless* versions of the graph matching problems in (Zanfir et al., 2018; Wang et al., 2019; 2021; 2020a; 2021; Jiang et al., 2022; Wang et al., 2020b; Fey et al., 2020; Rolínek et al., 2020; Gao et al., 2021; Yu et al., 2021c). For such seedless matching problems, non-topological node features are often assumed to be available. Thus, a node-based GNN is effective in learning how to extract useful node representations from high-quality non-topological node features. Unfortunately, from our own experience, we found that it is not easy to design a node-based GNN that effectively utilizes seed information (see

detailed discussions in Section 4.1). In contrast, our pair-wise SeedGNN architecture is much more effective in learning how to use seed information and synthesize various features. The design of our pair-wise architecture is also related to the work in Rolínek et al. (2020); Wang et al. (2021). However, Rolínek et al. (2020) still corresponds to a node-based GNN method and relies on high-quality node features. The NGM architecture of Wang et al. (2021) is most similar to us. However, NGM was not designed for seeded graph matching and has not been evaluated for this purpose either. See detailed discussion Appendix A.1 and our experiments in Section 5.2, where we find that, when extended to seeded graph matching, the NGM algorithm in Wang et al. (2021) does not generalize well when the test graph is with much larger size and node-degree than the training graph.

**Inductive Semi-supervised Learning on Graphs**  Our goal of using supervised learning for seeded graph matching shares some similarity with Wen et al. (2021), which also aims to both perform inductive learning (i.e., learn transferable knowledge from training graphs) and use a small amount of labeled data on the test graph. However, Wen et al. (2021) focuses on node classification, which is quite different from seeded graph matching. See detailed discussion in Appendix A.

More discussion on additional related work is deferred to Appendix A.

## 3  PROBLEM DEFINITION

We represent a graph of $n$ nodes by $\mathcal{G} = (V, \mathbf{A})$, where $V = \{1, 2, ..., n\}$ denotes the node set, and $\mathbf{A} \in \{0, 1\}^{n \times n}$ denotes the symmetric adjacent matrix, such that $\mathbf{A}(i, j) = 1$ if and only if nodes $i$ and $j$ are connected. For seeded graph matching, we are given two graphs $\mathcal{G}_1 = (V_1, \mathbf{A}_1)$ of $n_1$ nodes and $\mathcal{G}_2 = (V_2, \mathbf{A}_2)$ of $n_2$ nodes. Without loss of generality, we assume $n_1 \leq n_2$. There is an *unknown* injective mapping $\pi : V_1 \to V_2$ between $\mathcal{G}_1$ and $\mathcal{G}_2$. When $\pi(i) = j$, we say that $i \in V_1$ corresponds to $j \in V_2$. Throughout the paper, we denote a node-pair by $(i, j)$, where $i \in V_1$ and $j \in V_2$. For each node-pair $(i, j)$, if $j = \pi(i)$, then $(i, j)$ is a *true pair*; if $j \neq \pi(i)$, then $(i, j)$ is a *fake pair*. Then, an initial seed set $\mathcal{S}$ containing a fraction of true pairs is given. The goal of seeded graph matching is to recover the ground-truth mapping $\pi$ based on the observation of $\mathcal{G}_1$, $\mathcal{G}_2$ and $\mathcal{S}$.

In this work, we consider the problem of seeded graph matching in the supervised setting. The training set consists of several pairs of graphs, their initial seeds, and ground-truth mappings. Specifically, we use $\mathcal{T} = \{(P^{(1)}, \pi^{(1)}), (P^{(2)}, \pi^{(2)}), ..., (P^{(N)}, \pi^{(N)})\}$ to denote the training set, where $P^{(i)} = (\mathcal{G}_1^{(i)}, \mathcal{G}_2^{(i)}, \mathcal{S}^{(i)})$ denotes the $i$-th training example and $\pi^{(i)}$ is the ground-truth mapping for the $i$-th training example. For different training examples, the sizes of graphs and seed sets could be different. Our goal is to design a GNN architecture that can learn from training examples to predict the ground-truth mappings for unseen test graphs.

## 4  THE PROPOSED METHOD

In this section, we present in detail our proposed SeedGNN for seeded graph matching. See Figure 1 for a high-level illustration. In Section 4.1, we briefly illustrate the limitations of node-based GNNs for seeded graph matching, which motivate us to design pair-wise GNNs. We then describe the convolution module in Section 4.2, and the percolation module in Section 4.3. The supervised training procedure is presented in Section 4.4.

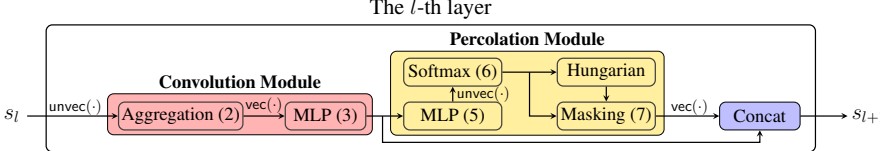

Figure 1: An overview of the $l$-th layer of our SeedGNN architecture. There are $L$ layers in total and each layer consists of two main modules. With the node-pair representations $s_l$ as input, the convolution module is a *local* processing step that aggregates the neighborhood information of each node-pair and updates the representation of its similarity through a neural network. The percolation module is a *global* processing step that compares the updated similarities of all node-pairs and finds the high-confidence ones. Then, we combine the local and global information from the two modules and propagate the new representations $s_{l+1}$ to the next layer.

**Notation**  we use vec($\cdot$) to denote the vectorization that converts a $n_1 \times n_2 \times d$ matrix to a matrix of $n_1 n_2 \times d$, where the $(i, j, :)$-th entry of the input matrix is the $((i-1)n_2 + j, :)$-th entry of the output matrix. Then, the unvectorization unvec($\cdot$) is the inverse operation of the vectorization.

## 4.1 FROM NODE-BASED GNNS TO PAIRWISE GNNS

Since most existing approaches for graph matching use node-based GNNs (Zhang et al., 2019; Chen et al., 2020; Wang et al., 2019; 2020a; Fey et al., 2020; Rolínek et al., 2020), we also started with node-based GNNs when we first researched the supervised approach to seeded-graph matching. However, from our experience, we found that node-based GNNs have significant difficulty effectively utilizing seed information. Note that node-based GNNs are adequate for the semi-supervised setting because the seed information is only used in the training *objectives*. However, as we discuss earlier in the introduction, this semi-supervised approach does not lead to transferable knowledge to unseen graphs. In contrast, to apply the node-based GNNs in the supervised setting, the first difficulty is to figure out a way to encode seed information as *input*. One natural attempt is to convert seeds to node features as input. For example, one could apply one-hot encoding, which represents the $i$-th seeds as a binary vector with the $i$-th element being 1 and 0 otherwise. However, this method needs to pre-specify the maximum number of seeds, and thus can not generalize to new graphs with even more seeds. Another possible method is random encoding, which uses a random vector to represent each seed. However, the vector dimension must also be chosen to be sufficiently large; otherwise two vectors corresponding to different seeds may have too similar encodings, which may confuse the GNN. This dependency on the vector dimension will again lead to generalization issue when the test graphs have much more seeds than the training graphs.

To avoid the generalization difficulty, an alternative approach is to use auxiliary "cross-links" across the two graphs to represent the seeds (see Figure 2). With these "cross-links", we can then combine the two graphs together and apply the node-based GNN on this union graph. This method is easier for generalization because the same GNN can be applied to graphs with arbitrary size and number of seeds. However, the topological structure of this union graph only informs the GNN that there is a seed at a particular location in the neighborhood, but not the seed identity. For example, in Figure 2, even though node 1 and node $4'$ have different seeds in their neighborhoods, their local neighborhood typologies (and the seed positions) look exactly the same. Thus, it would be difficult for a node-based GNN to produce node representations that can distinguish the two nodes.

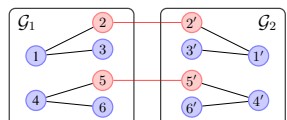

Figure 2: Let $i' = \pi(i)$. The red node-pairs are seeds. The red edges are cross-links.

To circumvent the aforementioned issues of existing node-based GNNs, we instead apply GNNs on node-pairs instead of nodes. Intuitively, when we apply such a pair-wise GNN to the node-pairs $(1, 1')$ and $(1, 4')$ in Figure 2, it can easily tell that $(1, 1')$ has a common seed, while $(1, 4')$ does not. As a result, this pair-wise GNN architecture will be able to learn generalizable knowledge from cross-links. We discuss the difference between our design and the previous pair-wise GNN architectures for seedless graph matching (Wang et al., 2021) in Appendix A.1. Below, we provide details of our proposed SeedGNN based on this idea.

## 4.2 CONVOLUTION MODULE

**Generalizable Encoding Method for Seeds.**  We follow the cross-link idea to encode the seed information as inputs for our SeedGNN, which is easier for generalization to unseen graphs as discussed earlier. Specifically, let $S_1 \in \{0, 1\}^{n_1 \times n_2}$ be the indicator matrix for seeds among $n_1 n_2$ node-pairs. If the node-pair $(i, j)$ is a seed, we let $S_1(i, j)$ be 1, and 0 otherwise. Then, we input $s_1 = \text{vec}(S_1) \in \{0, 1\}^{n_1 n_2 \times 1}$ into our SeedGNN.

With the input of seed information, we design the convolution module for SeedGNN that can count and utilize witnesses. We first introduce the definition of $l$-hop witnesses (Mossel et al., 2019). Given any graph $\mathcal{G}$ and two nodes $u, v$ in $\mathcal{G}$, we denote the length of the shortest path from $u$ to $v$ in $\mathcal{G}$ by $\text{dist}_\mathcal{G}(u, v)$. Then, for each node pair $(u, v)$ with $u$ in $\mathcal{G}_1$ and $v$ in $\mathcal{G}_2$, the seed $(w, \pi(w))$ becomes a $l$-hop witness for $(u, v)$ if $\text{dist}_{\mathcal{G}_1}(u, w) = l$ and $\text{dist}_{\mathcal{G}_2}(v, \pi(w)) = l$. Since we take the node-pair representations as input, our design is to apply SeedGNN to each node-pair across the two graphs. In this way, the pair-wise SeedGNN can directly infer from the neighboring topological

structure whether there is a witness. Specifically, taking the seed encoding vector $s_1$ as input, the counting of 1-hop witnesses can be written as

$$h_1 = (\mathbf{A}_1 \otimes \mathbf{A}_2)s_1,$$

where $\otimes$ denotes the Kronecker product. Likewise, we may further compute the $l$-hop witness-like information in the $l$-th layer of our SeedGNN as

$$h_l = (\mathbf{A}_1 \otimes \mathbf{A}_2)s_l, \tag{1}$$

where $s_l \in \mathbb{R}^{n_1 n_2 \times d_l}$ is specified later in Section 4.3, which contains the witness-like information within $(l-1)$-hops. Note that this expression (1) can be expanded as, for node-pair $(i, j)$,

$$h_l[(i-1)n_2 + j, :] = \sum_{(u,v):\, \mathbf{A}_1(u,i)=1,\, \mathbf{A}_2(v,j)=1} s_l[(u-1)n_2 + v, :],$$

which is similar to the aggregation step of the standard GNN in (Hamilton et al., 2017). The only difference is that we aggregate over a node-pair's neighborhoods. A direct implementation of (1) takes $O(n_1^2 n_2^2)$ computation, but we can reduce the complexity by letting $H_l = \mathsf{unvec}(h_l)$ and $S_l = \mathsf{unvec}(s_l)$, and rewriting (1) as

$$H_l[:,:,t] = \mathsf{unvec}((\mathbf{A}_1 \otimes \mathbf{A}_2)s_l[:,t]) = \mathbf{A}_1 S_l[:,:,t]\mathbf{A}_2, \quad t = 1, 2, ..., d_l. \tag{2}$$

Assume that the mean of the node degrees of $\mathcal{G}_1$ and $\mathcal{G}_2$ is $d_{\mathrm{mean}}$. Then, by sparse matrix multiplication, the complexity of the right-hand-side of (2) is reduced to $O(n_1 n_2 d_{\mathrm{mean}})$.

As we will see later in Section 4.3, in our SeedGNN $s_l$ will also contain outputs from the percolation layer. In order to learn how to best synthesize these two features (see further discussions in Section 4.3), we apply a neural network on $h_l$ after (1):

$$m_l = \phi_l(h_l), \tag{3}$$

where the update function $\phi_l$ is implemented as a $K$-layer neural network (we use $K = 2$ in our experiment). Let $\phi_l^{[0]}(h_l) = h_l$. The $k$-th layer of $\phi_l$ can be formulated as

$$\phi_l^{[k]}(h_l) = \sigma\left(\phi_l^{[k-1]}(h_l)\boldsymbol{W}^{[k-1]} + \boldsymbol{b}^{[k-1]}\right), \tag{4}$$

where $\boldsymbol{W}^{[k-1]}$ and $\boldsymbol{b}^{[k-1]}$ are learnable weights, initialized as Gaussian random variables; $\sigma$ is an activation function (we use ReLU). The updated representations $m_l \in \mathbb{R}^{n_1 n_2 \times (d_l - 1)}$ will be sent to the next layer of SeedGNN.

## 4.3 PERCOLATION MODULE

The percolation module is designed to match high-confidence nodes at one layer and to propagate the matched nodes as new seeds to the subsequent layers. Formally, we first obtain a similarity matrix in the $l$-th layer by mapping the node-pair representations $m_l$ to 1-dimension vectors, which is used to assess the similarity of each node-pair:

$$x_l = \rho_l(m_l). \tag{5}$$

We implement $\rho_l$ as multi-layer neural networks that is defined similarly as $\phi_l$ in (4). The output $x_l$ is in $\mathbb{R}^{n_1 n_2 \times 1}$. Then, we transform $x_l$ to $X_l = \mathsf{unvec}(x_l) \in \mathbb{R}^{n_1 \times n_2}$, and apply row-wise softmax to normalize $X_l$:

$$Y_l = \left(\mathrm{softmax}(X_l) + \mathrm{softmax}(X_l^\top)^\top\right)/2, \tag{6}$$

where for each row $\boldsymbol{v} = (v_1, ..., v_n) \in \mathbb{R}^n$ of input matrix, the softmax function is defined as

$$\mathrm{softmax}(\boldsymbol{v})_i = \frac{\exp(v_i)}{\sum_{j=1}^n \exp(v_j)}, \quad \text{for } i = 1, 2, .., n.$$

The similarity matrix $Y_l$ needs "cleaning" because it contains a lot of "noisy" information. For example, many fake pairs possess comparable similarity with true pairs (see Figure 3(a) for example). Further, there are far more fake pairs than true pairs. As a result, directly utilizing such misleading

information would lead to even more matching errors. Since the percolation idea from theoretical algorithms passes only new seeds with high confidence to the next stage (Yartseva et al., 2013), we leverage an approach called "masking" to remove the noisy information and retain the cleaner information in $Y_l$. More precisely, we utilize the Hungarian matching algorithm (Edmonds & Karp, 1972) to solve a linear assignment problem on $Y_l$ to find an injective mapping between $\mathcal{G}_1$ and $\mathcal{G}_2$, such that the total similarity of the matched node-pairs is maximized (see Figure 3(b) for example). The matching result is denoted by $R_l \in \{0, 1\}^{n_1 \times n_2}$, where $R_l(i, j) = 1$ if the node-pair $(i, j)$ is matched by the Hungarian algorithm and $R_l(i, j) = 0$ otherwise. Then, we filter out the noisy information in $Y_l$ by "masking":

$$z_l = \mathsf{vec}(Y_l \circ R_l), \tag{7}$$

where $\circ$ denotes element-wise multiplication (see Figure 3(c) for example). The matching information $z_l$ is sent to the next layer. As a result, many potentially noisy node-pairs are discarded. We note that both the idea of using similarity matrix to refine higher-layer matching and the idea of masking have appeared in seedless matching (Wang et al., 2019; Fey et al., 2020; Yu et al., 2019). However, there are crucial differences in the way that these ideas are utilized. Further, unlike previous percolation algorithms (Yartseva et al., 2013), our design of the percolation module can correct matching errors from earlier layers. We discuss these differences further in Appendix A.

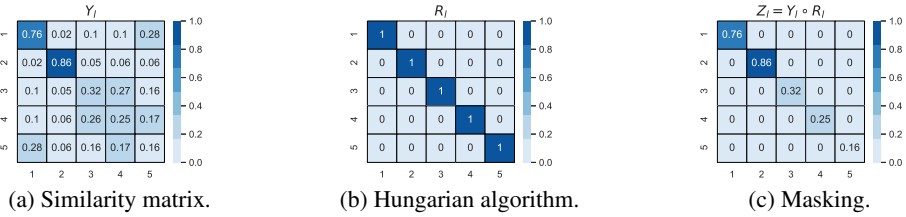

(a) Similarity matrix.     (b) Hungarian algorithm.     (c) Masking.

Figure 3: An illustration of our percolation process.

**The combination of the two features**    With the convolution module and the percolation module, our SeedGNN can identify witnesses at different hops and generate new seeds for percolation. However, these capabilities alone are insufficient. For example, when graphs are very sparse, even true node-pairs may not have enough witnesses if the number of hops $l$ is small. When graphs are very dense, a fake pair may also have many witnesses if $l$ is large. Thus, SeedGNN needs to learn how to adaptively utilize various types of witnesses in different types of graphs. Similarly, even with the above "cleaning" procedure, the output of the percolation module may still have low-confident seeds. Directly using them for percolation could lead to cascading errors. Thus, SeedGNN also needs to learn how to use new seeds with different levels of confidence.

The neural module in (3) is precisely designed to enable such learning. Specifically, instead of directly using the output $z_l$ from the percolation module as new seeds, we concatenate it with the output of the convolution module, i.e., $s_{l+1} = [m_l, z_l] \in \mathbb{R}^{n_1 n_2 \times d_{l+1}}$, as the input to the next layer. Then, after passing $s_{l+1}$ through (1), we apply the neural module (3). The joint effect of this design is that SeedGNN can utilize the confidence of $z_l$ to decide how much it should rely on various types of witnesses. Intuitively,

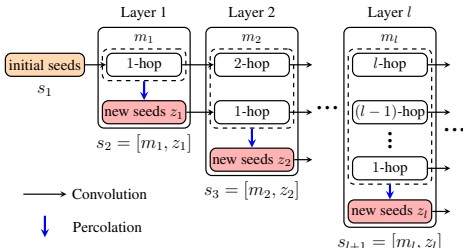

Figure 4: The witness information and new seeds computed by each layer.

at a higher layer $l \geq 2$, after passing $s_l$ by (1) and (3), $m_l$ may contain $l$-hop witness information from the initial seeds $s_1$, $(l-1)$-hop witness information from new seeds $z_1$, ... and 1-hop witnesses information from new seeds $z_{l-1}$ (see Figure 4). However, unlike the initial seeds $s_1$ that are either 0 or 1, the new seeds $z_1, z_2, ..., z_{l-1}$ also come with confidence levels. Thus, thanks to the non-linearity in $\phi_l$ at each layer, the strength of the various types of witness information (from either the initial seeds or the new seeds) will vary depending on the confidence levels of the new seeds, which then potentially allows SeedGNN to learn how to best utilize them adaptively. For example, for sparse graphs, the confidence levels of the new seeds in the first several layers are low. As a result, SeedGNN can utilize witnesses based on the initial seeds but at a larger number of hops. In contrast, for dense graphs, if the confidence levels of the new seeds in the first several layers are

already high, SeedGNN can then utilize the new witnesses computed from those new seeds. This capability is experimentally validated in Appendix C.3 by studying the layer-wise matching process of SeedGNN for different types of graphs. Further, SeedGNN can even combine different types of witness information together and potentially extract unknown but more valuable features.

### 4.4 Loss Function

Finally, we utilize the ground-truth node-to-node correspondence as the supervised training information for end-to-end training. More precisely, for any training example $(P, \pi) \in \mathcal{T}$, we adopt the cross-entropy loss to measure the difference between our prediction and the ground-truth mapping $\pi$. Then, we add up the cross-entropy loss of every layer:

$$
\mathcal{L}_P(\vartheta) = -\sum_{l=1}^{L} \left( \sum_{(i,j):\, j=\pi(i)} \log\left(Y_l(i,j) + \epsilon\right) + \sum_{(i,j),\, j\neq\pi(i)} \log\left(1 - Y_l(i,j) + \epsilon\right) \right),
$$

where $Y_l$ is given in (6), $\vartheta$ denotes all the learnable weights in the networks $\phi_l$ and $\rho_l$, and $\epsilon$ is a small positive value (e.g. $\epsilon = 10^{-9}$) to avoid a logarithm of zero. The total loss function is $\mathcal{L}(\vartheta) = \sum_{P\in\mathcal{T}} \mathcal{L}_P(\vartheta)$. We find that the use of the losses from all layers in training helps to speed up the training process. This is somewhat inspired by hierarchical learning methods in (Bengio, 2009; Schmidhuber, 1992; Simonyan & Zisserman, 2015). It allows the lower layers to be trained first, making it easier to train the next layers. Then, in testing, we will apply the trained SeedGNN model on the test graphs and only use the matching result of the final layer, $R_L$, as the predicted mapping since the final layer already synthesizes all the features learned at the lower layers.

The total time complexity of SeedGNN is $O(n_1 n_2^2)$, and the space complexity is $O(n_1 n_2)$. The detailed discussion on the complexity and scalability of our SeedGNN is deferred to Appendix B.

## 5 Experiments

In this section, we conduct numerical experiments to demonstrate the advantages of SeedGNN.

### 5.1 Experimental Set-up

The number of SeedGNN layers is empirically fixed to 6 throughout our experiments. We implement the operators $\phi_l$ and $\rho_l$ as two-layer neural networks with $d_l = 16$. For all experiments, optimization is done via ADAM (Kingma et al., 2015) with a fixed learning rate of $10^{-2}$. Our code is implemented using PyTorch (Paszke et al., 2019) and trained on an Intel Core i7-8750H CPU. The performance is evaluated using the matching accuracy rate, i.e., the fraction of nodes that are correctly matched.

**Datasets.** We use the correlated Erdős-Rényi graph model (Pedarsani et al., 2011), SHREC'16 dataset in (Lähner et al., 2016) and Facebook networks provided in (Traud et al., 2012) in our experiments. **1) The correlated Erdős-Rényi graph model**. We first generate the parent graph $\mathcal{G}_0$ from the Erdős-Rényi model $\mathcal{G}(n, p)$, i.e., we start with an empty graph on $n$ nodes and connect any pair of two nodes independently with probability $p$. Then, we obtain a subgraph $\mathcal{G}_1$ by sampling each edge of $\mathcal{G}_0$ into $\mathcal{G}_1$ independently with probability $s$. Repeat the same sub-sampling process independently and relabel the nodes according to a uniformly random permutation permutation $\pi$ to construct another subgraph $\mathcal{G}_2$. Then, each true pair is independently added into the seed set $\mathcal{S}$ with probability $\theta$. **2) Facebook networks.** The dataset in (Traud et al., 2012) provides 100 Facebook networks from different institutions. We randomly choose 10 for training and 90 for testing. The sizes of the Facebook networks range from 962 to 32361. To lower the training cost, we down-sample the sizes of the training graphs. Specifically, for each Facebook network for training, we first down-sample nodes with probability 0.25 to get the parent graph $\mathcal{G}_0$. However, for testing, we do not perform this down-sampling and use the original graphs directly as the parent graph $\mathcal{G}_0$. For both training and testing, we generate $\mathcal{G}_1$ and $\mathcal{G}_2$ from $\mathcal{G}_0$ by independently sub-sampling each edge of $\mathcal{G}_0$ twice with probability $s = 0.8$ and sub-sampling each node of $\mathcal{G}_0$ twice with probability 0.9. The nodes of $\mathcal{G}_2$ are then relabeled according to a random permutation $\pi$. Then, each true pair is independently added into the seed set $\mathcal{S}$ with probability $\theta$. **3) The SHREC'16 dataset**. Matching 3D deformable shapes is a central problem in computer vision, and has been extensively studied

for decades (see (Van Kaick et al., 2011) and (Sahillioğlu, 2020) for surveys). The SHREC'16 dataset in (Lähner et al., 2016) provides 25 deformable 3D shapes (15 for training and 10 for testing) undergoing different topological changes. Each shape is represented by a triangulated mesh graph consisting of around 8K-11K nodes (with 3D coordinates).

**Training set.** We construct the training set $\mathcal{T}$ in the following way. First, we generate 100 random pairs of correlated Erdős-Rényi graphs with $n = 100$, $p \in \{0.1, 0.3, 0.5\}$, $s \in \{0.6, 0.8, 1\}$, and $\theta = 0.1$. Second, we add 10 pairs of Facebook networks as discussed above with $\theta = 0.1$ into the training set. Third, we do not include any SHREC'16 dataset in the training set, because our SeedGNN trained on the above two datasets already performs well for the SHREC'16 dataset (see Section 5.2), which verifies the generalization power of our SeedGNN.

**Baselines.** We compare the performance of our proposed SeedGNN with several state-of-the-art algorithms: **1) $D$-hop** (Mossel et al., 2019) finds the node mapping between the two graphs that maximizes the total number of $D$-hop witnesses for a given $D$. For a fair comparison with other algorithms, we iteratively apply the $D$-hop algorithm $T$ times (with $DT = 6$ because SeedGNN is fixed to have 6 layers). In each iteration, we use the matching result of the previous iteration as new seeds and apply the $D$-hop algorithm again. **2) PGM** (Kazemi et al., 2015) iteratively matches node-pairs with at least $r$ witnesses. We choose $r = 2$, which is the same as the simulation setting in (Kazemi et al., 2015). **3) PLD** (Yu et al., 2021b) is the state-of-the-art seeded graph matching algorithm designed for graphs with power-law degree distributions (which is a common feature of real-world social networks (Barabási, 2016)). **4) SGM** (Fishkind et al., 2019) uses Frank–Wolfe method to approximately solves a quadratic assignment problem that maximizes the number of matched edges between two graphs, while being consistent with the given seeds. **5) MGCN** (Chen et al., 2020) is a representative semi-supervised learning-based GNN approach, whose performance is comparable with other semi-supervised learning approaches. The parameters are set in the same way as those in (Chen et al., 2020). **6) NGM** (Wang et al., 2021) is a supervised GNN method for seedless graph matching, but it also uses a pair-wise GNN that utilizes an affinity matrix as input. We transfer this approach to seeded graph matching by modifying the affinity matrix to also encode seed information. We then train the weights of NGM with the same training set as our SeedGNN.

## 5.2 RESULTS

**Performance Comparison on Correlated Erdős-Rényi Model.** In Figure 5, we show the performance of the algorithms on the correlated Erdős-Rényi graph model. For test graphs, we vary $\theta$ while fixing $n = 500$, $p \in \{0.01, 0.2\}$, $s = 0.8$. The different graph sizes between the training set ($n = 100$) and the test set ($n = 500$) aim to demonstrate the generalization power of our SeedGNN. We can observe that, among the state-of-the-art methods, the iterative 2-hop algorithm has the best performance for sparse graphs ($p = 0.01$), and the SGM algorithm performs the best for dense graphs ($p = 0.2$). However, our SeedGNN has overall the best performance among all algorithms. From theoretical graph matching results (Mossel et al., 2019), we have learned that we need to use witnesses at different numbers of hops for matching sparse graphs ($p = 0.01$) and dense graphs ($p = 0.2$). Thus, these results suggest that our SeedGNN chooses the appropriate features to match different types of graphs. Further, from the fact that our SeedGNN outperforms existing algorithms even in the settings that they work well, it indicates that our SeedGNN is able to utilize the features more effectively than the state-of-the-art methods. When comparing the performance of SeedGNN with NGM (which also uses a pair-wise GNN architecture), we can observe from Figure 5 that, although NGM performs close to our SeedGNN in larger sparse graphs ($p = 0.01$), it performs quite poorly in larger dense graphs ($p = 0.2$). We discuss the possible reason in Appendix A.1.

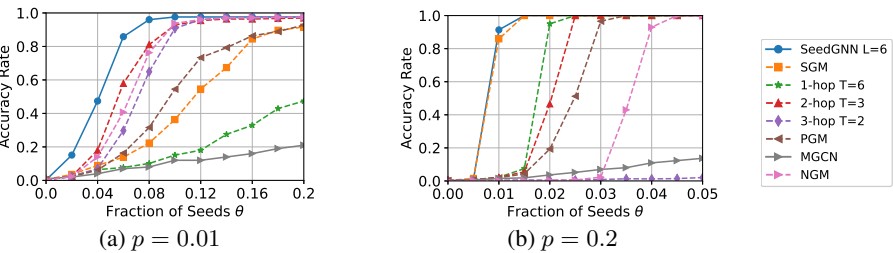

(a) $p = 0.01$          (b) $p = 0.2$

Figure 5: Performance comparison on correlated Erdős-Rényi graphs. Fix $n = 500$ and $s = 0.8$.

**Performance Comparison on Facebook Networks.** We compare the SeedGNN with the state-of-the-art algorithms on the 90 Facebook networks for testing, when we vary $\theta$ from 0 to 0.05. We show the performance of these algorithms in Figure 6. We can observe that our SeedGNN is comparable with SGM and significantly outperforms other algorithms. Note that the matching accuracy is saturated at around $80\%$, because there are about $15\%$ nodes that do not have any common neighbour in $\mathcal{G}_1$ and $\mathcal{G}_2$, and thus can not be correctly matched.

**Performance Comparison on SHREC'16 Dataset.** To validate that our SeedGNN can adapt to different graphs, we evaluate SeedGNN for deformable shape matching using the SHREC'16 dataset. Note that the sizes and types of graphs in this dataset are quite different from the Erdős-Rényi and Facebook graphs in the training set. We can see in Figure 7 that our SeedGNN still outperforms all baselines. This improvement demonstrates that SeedGNN trained with the other two datasets can generalize to real-world graphs with different sizes and types. We note that MGCN (a GNN based on semi-supervised training) utilizes both topological structures and non-topological node features. However, in this application the non-topological node features correspond to 3D coordinates, which do not provide much useful information for correlating two 3D shapes with different poses. As a result, MGCN algorithm using 3D coordinates as node features almost fails completely. To confirm that the non-topological node features in the SHREC'16 dataset are not very helpful, we also run supervised learning methods for seedless graph matching that only rely on the 3D coordinates. As we shown in Table 1, they all suffer poor performance. Due to this reason, MGCN does not perform well either. To further validate that our SeedGNN use seed information more effectively than other supervised learning methods, we use the random encoding mentioned in Section 4.1 to represent seed information, and provide them as input for supervised GNNs (except NGM, for which we directly modify the affinity matrix using seed information). We fix the fraction of seeds $\theta$ at 0.01 and the encoding vector dimension at 16. In Table 1, we can observe that both our SeedGNN and NGM outperform the supervised methods using node-based GNNs, even when the latter are augmented with seed information. This suggests that our method is more effective in using seed information than the node-based supervised GNN methods. Moreover, we discuss why NGM has a similar performance with our SeedGNN on the SHREC'16 dataset in Appendix A.1. In Table 1, we also show the average run time of supervised learning algorithms to match a pair of graphs. We can observe that our SeedGNN is comparable with these algorithms on large graphs. Additional numerical studies to compare SeedGNN with more state-of-the-art GNN approaches on another real graph datasets with different sizes and types are deferred to Appendix D.

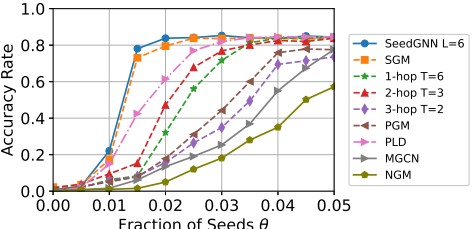

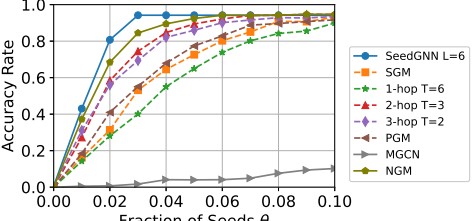

Figure 6: Performance comparison on the Facebook networks with different $\theta$.

Figure 7: Performance comparison on the SHREC'16 dataset with different $\theta$.

Table 1: Comparison of GNN methods on SHREC'16 dataset. The best results are marked as bold.

| Method | | Semi-Supervised | | | Supervised | | | | SeedGNN |
|---|---|---|---|---|---|---|---|---|---|
| | | DeepLink Zhou et al. (2018) | CrossMNA Chu et al. (2019) | MGCN Chen et al. (2020) | DGMC Fey et al. (2020) | BB-GM Rolínek et al. (2020) | DGM Gao et al. (2021) | NGM Wang et al. (2021) | ours |
| Accuracy | seeded | $3.3 \pm 0.8$ | $4.2 \pm 1.7$ | $3.8 \pm 1.1$ | $23.2 \pm 6.8$ | $21.1 \pm 4.4$ | $19.31 \pm 10.6$ | $37.9 \pm 5.7$ | $\mathbf{43.1 \pm 8.5}$ |
| (%) | seedless | – | – | – | $0.1 \pm 0.0$ | $0.1 \pm 0.0$ | $0.1 \pm 0.0$ | $0.1 \pm 0.0$ | – |
| run time (s) | | – | – | – | 80.2 | 130.1 | 211.3 | 879.2 | 141.5 |

**Additional experiments.** We conduct additional experiments to further investigate the inner working of our SeedGNN (please refer to Appendix C for details). First, to verify the effectiveness of our design choices for SeedGNN, we compare the performance of different architectural designs. Then, we investigate which sets of samples need to be included in our training set to obtain an effective trained model. Finally, we study the matching process of SeedGNN for different types of graphs. The results suggest that SeedGNN chooses the appropriate features for different graphs based on the confidence level of new seeds.

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

## A    ADDITIONAL RELATED WORK

**Theoretical Algorithms**    Existing theoretical algorithms suffer from several limitations. First, graphs with different characteristics (e.g., Erdős-Rényi graphs versus power-law graphs) may require different features and carefully tuned parameters. In contrast, by learning from the training graphs, our SeedGNN can automatically choose the effective features. Second, these theoretical algorithms may not synthesize different features most effectively. For instance, the $l$-hop algorithm in (Mossel et al., 2019) only utilizes the witnesses at a specific hop $l$, but does not study how to combine witnesses at different hops.

**Inductive Semi-supervised Learning on Graphs**    Our goal of using supervised learning for seeded graph matching shares some similarity with the work in Wen et al. (2021), which also aims to both perform inductive learning (i.e., learn transferrable knowledge from training graphs) and utilize a small amount of labeled data on the test graph. However, Wen et al. (2021) focuses on a node classification problem, which is quite different from seeded graph matching. In particular, Wen et al. (2021) uses node-based GNNs, which (as we discussed in Section 4.1) have more difficulty in effectively utilizing seed information than our proposed pair-wise GNN. Further, in order to transfer knowledge from the trained GNN to test graphs, Wen et al. (2021) scales all GNN weights by a common factor. It is unclear how this scaling will effectively transfer knowledge for seeded graph matching, e.g., how to best use different hops of witnesses. In contrast, our design of SeedGNN exploits the inherent structure of the seeded graph matching problem, and can be shown to generalize well to unseen graphs of sizes and types very different from the training set. For future work, it would be of interest to explore whether our SeedGNN can be further improved with a meta-learning component (Santoro et al., 2016).

**Convex Relaxation Algorithms**    In addition to the theoretical algorithms and the GNN approaches, there is another class of algorithms based on convex relaxations of the quadratic assignment problem, which maximizes the total number of matched edges between two graphs subject to the seed constraint (Lyzinski et al., 2014; Fishkind et al., 2019). In (Fishkind et al., 2019), the authors describe a gradient ascent approach to solve this relaxed problem, which is called SGM. Compared to SeedGNN, SGM also has flavors of using witnesses and percolation ideas. Specifically, the gradient of the SGM algorithm is similar to a matrix counting 1-hop witnesses. However, using only 1-hop witnesses is known to be ineffective in sparse graphs (as there are very few 1-hop witnesses even for true pairs). Indeed, our experiments in Section 5 find that our SeedGNN often outperforms SGM, especially in sparse graphs.

**Differences in Using Similarity Matrix and Masking**    We note that both the idea of using similarity matrix to refine higher-layer matching and the idea of masking have appeared in seedless matching. For example, some previously proposed node-based GNN architectures for *seedless* graph matching also compute the similarity matrix and use it to refine the node embedding in each layer (Wang et al., 2019; Fey et al., 2020). However, these approaches heavily rely on high-quality non-topological node features and does not clean up the "noisy" information as we carefully did. Yu et al. (2019) also uses the Hungarian algorithm for seedless graph matching, but they only clean up the matching result in their loss function. The results of intermediate layers are still very noisy. We use the Hungarian algorithm in each layer to filter out the misleading information, and thus the final result would be better.

**Differences between Our Percolation Module and Previous Percolation Aalgorithms**    Unlike previous percolation algorithms (Yartseva et al., 2013), we allow SeedGNN to correct errors from ealier layers by re-matching nodes at each layer. Note that in many percolation algorithms, once a new pair of seeds is identified, it will be used as the correct matching until the end. This approach can be problematic if an incorrect pair is identified as seeds, whose impact will be lasting for many iterations down the road. In contrast, since our SeedGNN rematches nodes at each layer, even if some of the newly-identified seeds in the previous layer are incorrect, we can potentially correct these errors in the next layer, as long as the fraction of incorrect seeds is small. In other words, our design of SeedGNN takes advantage of the power of partially-correct (i.e., noisy) seeds (as theoretically verified in Yu et al. (2021a))

A.1 COMPARISON WITH NGM

The NGM architecture of Wang et al. (2021) shares some similarity with our SeedGNN, and it also uses a pair-wise GNN and uses the affinity matrix as input. However, note that Wang et al. (2021) focuses on seedless graph matching. Therefore, the NGM architecture in Wang et al. (2021) was not designed for seeded graph matching. For example, they do not aim to exploit important features such as witness. Further, the NGM algorithm has not been evaluated for seeded graph matching either. In Section 5.2, we transferred the NGM approach to seeded graph matching by modifying the affinity matrix to encode seed information, and compare its performance with our SeedGNN. Through these experiments, we find that the NGM algorithm in Wang et al. (2021) does not generalize well when the test graph is with much larger size and node-degree than the training graph. Below, we discuss the possible reasons.

Recall that we train both NGM and our SeedGNN on the same training set in Section 5.1, and test on correlated Erdős-Rényi graphs with $n = 500, s = 0.8, p = \{0.01, 0.2\}$. Note that this test graph size is larger than the training Erdős-Rényi graph size of $n = 100$. From the experimental results in Figure 5 in Section 5.2, we can observe that, although NGM performs close to our SeedGNN in larger sparse graphs ($p = 0.01$), it performs quite poorly in larger dense graphs ($p = 0.2$). One possible reason for this deterioration in the generalization power of NGM could be that, in the aggregation step, NGM normalizes each representation by the vertex degree of the association graph (which is roughly the square of the node degrees), but we do not. To see why this difference matters, note that according to known theoretical results on seeded graph matching, there exist algorithms that only need $\Omega(\log n)$ seeds to match all $n$ nodes (Mossel et al., 2019). However, if the graph sparsity $p$ is fixed, the node degree increases proportionally to $n$, and correspondingly the vertex degree of the association graph increases quadratically with $n$. As a result, when NGM divides the similarity of each node pair by the vertex degree, we expect that the resulting value ($\sim \frac{\log n}{n^2}$) will decrease close to zero as the graph size increases. Hence, it would be difficult for the sinkhorn step in NGM to distinguish the true pairs from the fake pairs in test graphs with larger size and node degrees than the training graphs. In contrast, since SeedGNN does not divide the similarity scores by the vertex degrees, the Hungarian algorithm step in our percolation (which can distinguish any absolute difference) will then be able to distinguish the true pairs from the fake pairs.

In contrast to Figure 5(b), for the experiment on the SHREC'16 dataset (Table 1), NGM has similar performance as our SeedGNN. This is because in this experiment, we train NGM also with the SHREC '16 dataset (same as other seedless GNNs in Table 1). Note that the node degrees of the graphs in the SHREC'16 dataset are all around 6. In other words, the test graphs and training graphs are with similar node degrees. As a result, the issue caused by dividing the similarity scores by the vertex degree of the associate graphs is not as critical for the SHREC'16 dataset.

# B COMPLEXITY AND SCALABILITY

## B.1 TIME AND SPACE COMPLEXITY

First, we analyze the computational complexity of our SeedGNN. In each layer, counting witnesses in (2) takes $O(n_1 n_2 d_{\text{mean}})$ time. The neural networks (3) and (5) take $O(n_1 n_2)$ time. The Hungarian algorithm takes $O(n_1 n_2^2)$ times (Crouse, 2016). Thus, the total time complexity is $O(n_1 n_2^2)$.

The space complexity of our SeedGNN is $O(n_1 n_2)$ since we need to store the representations of all $n_1 n_2$ node-pairs in each layer.

## B.2 MAKING SEEDGNN MORE SCALABLE

For very large graphs, the step of the Hungarian algorithm may potentially become the computational bottleneck. We can use greedy max-weight matching (GMWM) in (Avis, 1983) instead, as the time complexity of GMWM is only $O(n_1 n_2 \log n_2)$. With this improvement, the total time-complexity is reduced to $O(n_1 n_2 \log n_2 + n_1 n_2 d_{\text{mean}})$. To the best of our knowledge, the best-known time complexity for GNN-based algorithms is $O(n_1 n_2)$ (Fey et al., 2020). Thus, the computational complexity of our SeedGNN is only moderately larger than the best known one. Our numerical

result shown in Table 1 has demonstrated that the run time of our SeedGNN is comparable to the best-known GNN-based algorithms.

## C   STUDYING THE INNER-WORKING OF SEEDGNN

In this section, we further investigate how the performance of SeedGNN varies as we change its inner working. First, to verify the effectiveness of our design choices for our SeedGNN method, we compare the performance of different architectural designs. Then, we investigate which sets of samples need to be included in our training set to obtain an effective trained model. Finally, we study the matching process of SeedGNN for different types of graphs. The results suggest that SeedGNN could potentially choose the appropriate features for different graphs based on the confidence level of new seeds.

### C.1   STUDY OF THE DESIGN CHOICES

To verify the effectiveness of our design choices, we consider four variants of SeedGNN, which are:

1. **SeedGNN-x:** SeedGNN without convolution module. This variant aims to verify the importance of extracting witness-like information at a larger number of hops.

2. **SeedGNN-w:** SeedGNN without percolation module. This variant aims to verify the importance of the percolation module in SeedGNN.

3. **SeedGNN-p:** SeedGNN with percolation module but without the Hungarian matching algorithm (i.e., $z_l = \mathsf{unvec}(Y_l)$ in each GNN layer). This variant aims to verify the importance of the "cleaning" process in SeedGNN.

4. **SeedGNN-h:** SeedGNN with $z_l = \mathsf{unvec}(R_l)$ instead of (7) in each layer. This variant aims to verify that among the new seeds, it is still important to distinguish the high-confident one and low-confident one.

Finally, we use "SeedGNN" to denote the full design in Fig. 1. We train all these variants with the same training set $\mathcal{T}$ in Section 5.1.

In Figure 8, we show the performance of the above variants of SeedGNN on correlated Erdős-Rényi graph model. For test graphs, we increase $\theta$ from 0 to 0.05 while fixing $n = 500$, $p = 0.04$, $s = 0.8$. As illustrated in Figure 8, our SeedGNN with full design achieves the best performance among all variants, which shows the effectiveness of our design choices for the SeedGNN architecture. Further, among the variants, SeedGNN-w almost fails completely, which highlights the significant importance of using the percolation idea in SeedGNN for seeded graph matching. SeedGNNx does performs poorly, which demonstrates that it is also important to extract witness information at a larger number of hops instead of only 1-hop. We can observe that SeedGNN and SeedGNN-h both outperform SeedGNN-p and the improvement of SeedGNN is significantly bigger. This result verifies that it is not enough to only use the soft-correspondence (as in SeedGNN-p), and we need to combine both the matching result $R_l$ of the Hungarian algorithm and the similarity $Y_l$ as in (7) to achieve the best performance.

### C.2   STUDY OF THE NECESSARY TRAINING SAMPLES FOR GENERALIZATION

Intuitively, in order to help our SeedGNN successfully learn useful knowledge that can be applied to never-seen graphs, the training set needs to contain graph pairs with different varieties, e.g., graph sparsity, graph correlation, and the size of seed set. However, a larger training set also increases the training time. To show which sets of graph pairs are necessary, we compare SeedGNN trained with different training sets, whose parameters are shown in Table 2. We use $\mathcal{T}$ to denote the training set that only includes the Erdős-Rényi graphs of the training set in Section 5.1. First, to show the necessity of training graph pairs with a wide range of sparsity, we train SeedGNN with $\mathcal{T}$, $\mathcal{T}_{p1}$ and $\mathcal{T}_{p2}$, and compare the performance of the trained models while increasing $p$ from 0.02 to 0.2 and fixing $n = 500$, $s = 0.8$ and $\theta = 0.05$. Figure 9(a) shows that, if SeedGNN is only trained with $p = 0.1$, it performs well on sparse graphs but poorly on dense graphs. In contrast, if SeedGNN is only trained with $p = 0.5$, it performs well on dense graphs but poorly on sparse graphs. Thus, we should include both $p = 0.1$ and $p = 0.5$ in the training set to achieve good performance. Second, to

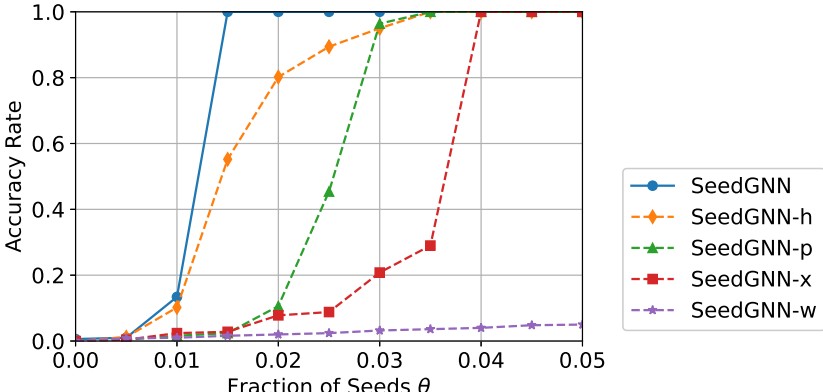

Figure 8: Performance comparison of our SeedGNN and four other variants on correlated Erdős-Rényi graph model with different $\theta$. Fix $n = 500$, $p = 0.04$, $s = 0.8$.

show the necessity of training graph pairs with different correlations, we compare the performance of SeedGNN trained with $\mathcal{T}$, $\mathcal{T}_{s1}$, $\mathcal{T}_{s2}$ and $\mathcal{T}_{s3}$, and compare these models while increasing $s$ from $0.5$ to $1$ and fixing $n = 500$, $p = 0.08$ and $\theta = 0.05$. Figure 9(b) shows that, if SeedGNN is only trained with $s = 0.6$, it performs well on moderately correlated graphs but poorly on highly correlated graphs. In contrast, if SeedGNN is only trained with $s = 0.8$ or $s = 1$, it performs well on highly correlated graphs but poorly on moderately correlated graphs. Thus, we should include different correlations in the training set to achieve good performance. Third, we compare the performance of SeedGNN trained with $\mathcal{T}$, $\mathcal{T}_{t1}$ and $\mathcal{T}_{t2}$, and compare these models while increasing $\theta$ from $0$ to $0.05$ and fixing $n = 500$, $p = 0.04$ and $s = 0.8$. Figure 9(c) shows that, if SeedGNN is only trained with $\theta = 0.1$ and $\theta \in \{0.1, 0.3\}$, it performs exactly the same. If SeedGNN is only trained with $\theta = 0.3$, it performs worse than the former two. Thus, we only need to include graph pairs with a relatively small seed set in the training set.

Table 2: Different Training Sets

| Training Sets | $p$ | $s$ | $\theta$ |
|---|---|---|---|
| $\mathcal{T}_{p1}$ | $\{0.1\}$ | $\{0.6, 0.8, 1\}$ | $\{0.05, 0.1\}$ |
| $\mathcal{T}_{p2}$ | $\{0.5\}$ | $\{0.6, 0.8, 1\}$ | $\{0.05, 0.1\}$ |
| $\mathcal{T}_{s1}$ | $\{0.1, 0.5\}$ | $\{1\}$ | $\{0.05, 0.1\}$ |
| $\mathcal{T}_{s2}$ | $\{0.1, 0.5\}$ | $\{0.8\}$ | $\{0.05, 0.1\}$ |
| $\mathcal{T}_{s3}$ | $\{0.1, 0.5\}$ | $\{0.6\}$ | $\{0.05, 0.1\}$ |
| $\mathcal{T}_{t1}$ | $\{0.1, 0.5\}$ | $\{0.6, 0.8, 1\}$ | $\{0.3\}$ |
| $\mathcal{T}_{t2}$ | $\{0.1, 0.5\}$ | $\{0.6, 0.8, 1\}$ | $\{0.1, 0.3\}$ |

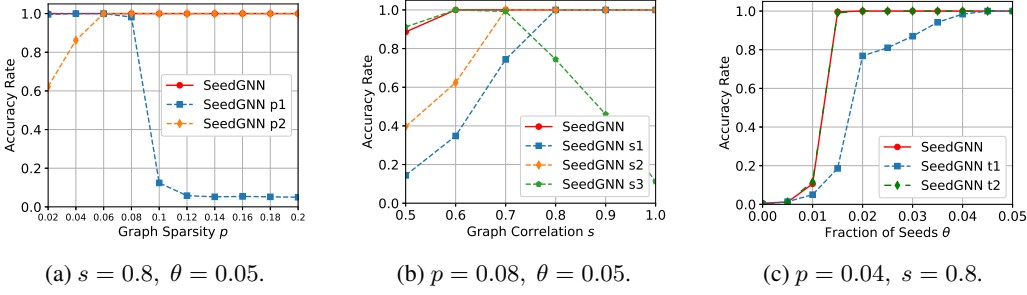

(a) $s = 0.8$, $\theta = 0.05$.  (b) $p = 0.08$, $\theta = 0.05$.  (c) $p = 0.04$, $s = 0.8$.

Figure 9: Performance comparison of SeedGNN trained with different training sets. Fix $n = 500$.

## C.3 LAYER-WISE STUDY OF SEEDGNN

Recall from Section 4.3 that our design on the feature combination potentially enables SeedGNN to utilize various types of witness information adaptively, based on the confidence levels of new seeds $z_l$. In this section, we verify this capability through numerical results. To directly visualize $z_l$ in the matching process, we present the similarity matrix $Y_l$ of each layer of SeedGNN and compare it with the witness matrix of the iterative 1-hop and 2-hop algorithms at each iteration. We assume that the true mapping $\pi$ is the identity permutation, i.e., $\pi(i) = i$.

First, we study the matching process in *dense* graphs. We fix a pair of correlated Erdős-Rényi graphs with $n = 50$, $p = 0.4$, $s = 0.8$ and $\theta = 0.1$. Then, we index the nodes from 0 to 49 in the descending order of the node degree in the parent graph $\mathcal{G}_0$. In Figure 10, we show the similarity matrix $Y_l$ in each layer of our SeedGNN, and compare it with the witness matrix in each iteration using either the 1-hop or 2-hop algorithm. We can immediately see that the similarity matrices provided by SeedGNN are more similar to the witness matrices of the iterative 1-hop algorithm than that of the iterative 2-hop algorithm. Specifically, since the graphs are dense, the 1-hop witness information from the initial seeds can already generate new seeds with high confidence (see Figure 10(a) and 10(g), where there are many dark points on the diagonal (i.e., consistent with the underlying true mapping), while there are few dark points off the diagonal). The iterative 1-hop algorithm is known to use new 1-hop witnesses from these new seeds (see Figure 10(h)) in the next iteration. In contrast, the 2-hop witnesses from the initial seeds are much noisier (see Figure 10(m), where the darkness of the points on the diagonal cannot be differentiated from those off the diagonal). As we illustrated in Figure 4, these two types of witness information are both contained in the second layer of SeedGNN. By comparing Figure 10(b) with Figure 10(h) and Figure 10(m), we can observe that the second layer of SeedGNN produces a similarity matrix that is closer to the witness matrix of the 1-hop algorithm than that of the 2-hop algorithm. Thus, we infer that, for these dense graphs in which the new seeds are reliable, the SeedGNN relies more on witnesses computed from these new seeds.

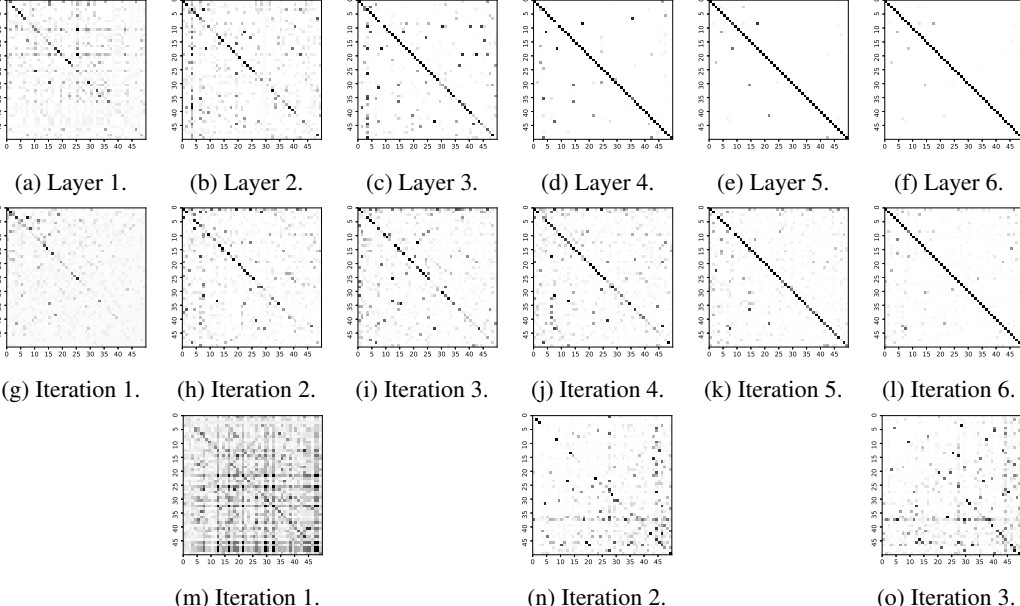

|  |  |  |
|---|---|---|
| (a) Layer 1. | (b) Layer 2. | (c) Layer 3. | (d) Layer 4. | (e) Layer 5. | (f) Layer 6. |

| (g) Iteration 1. | (h) Iteration 2. | (i) Iteration 3. | (j) Iteration 4. | (k) Iteration 5. | (l) Iteration 6. |

| (m) Iteration 1. | (n) Iteration 2. | (o) Iteration 3. |

Figure 10: The similarity/witness matrices of the matching process on a fixed pair of dense correlated Erdős-Rényi graphs with $n = 50$, $p = 0.4$, $s = 0.8$ and $\theta = 0.1$. Darker points correspond to higher similarity (in $Y_l$) or a larger number of witnesses. Figure 10(a) — Figure 10(f) are the similarity matrix from each layer of SeedGNN. Figure 10(g) — Figure 10(l) are the witness matrix from each iteration of the iterative 1-hop algorithm. Figure 10(m) — Figure 10(o) are the witness matrix from each iteration of the iterative 2-hop algorithm.

Then, we study the matching process in *sparse* graphs. We fix a pair of correlated Erdős-Rényi graphs with $n = 50$, $p = 0.1$, $s = 0.8$ and $\theta = 0.1$. Then, we also index the nodes from 0 to 49 in

the descending order of the node degree in the parent graph $\mathcal{G}_0$. In Figure 11, we show the similarity matrix $Y_l$ in each layer of our SeedGNN, and compare it with the witness matrix in each iteration using either the 1-hop or 2-hop algorithm. In contrast to Figure 10, in this case, we observe that the similarity matrices provided by SeedGNN are more similar to the witness matrices of the iterative 2-hop algorithm than those of the iterative 1-hop algorithm. Specifically, since the graphs are sparse, there are very few 1-hop witnesses even for true pairs. Thus, the 1-hop algorithm almost fails completely (see Figure 11(g) — Figure 11(l)). On the contrary, the 2-hop witnesses from the initial seeds are much more reliable (see Figure 11(m)). As a result, the iterative 2-hop algorithm produces much better results (see Figure 11(m) — Figure 11(o)). By comparing Figure 11(b) with Figure 11(h) and Figure 11(m), we can observe that the second layer of SeedGNN produces a similarity matrix that is closer to the witness matrix of the 2-hop algorithm than that of the 1-hop algorithm. Thus, we can infer that, for these sparse graphs in which the confidence levels of new seeds are low, SeedGNN utilizes 2-hop witness information from the initial seeds, and avoids using 1-hop witnesses based on these new seeds.

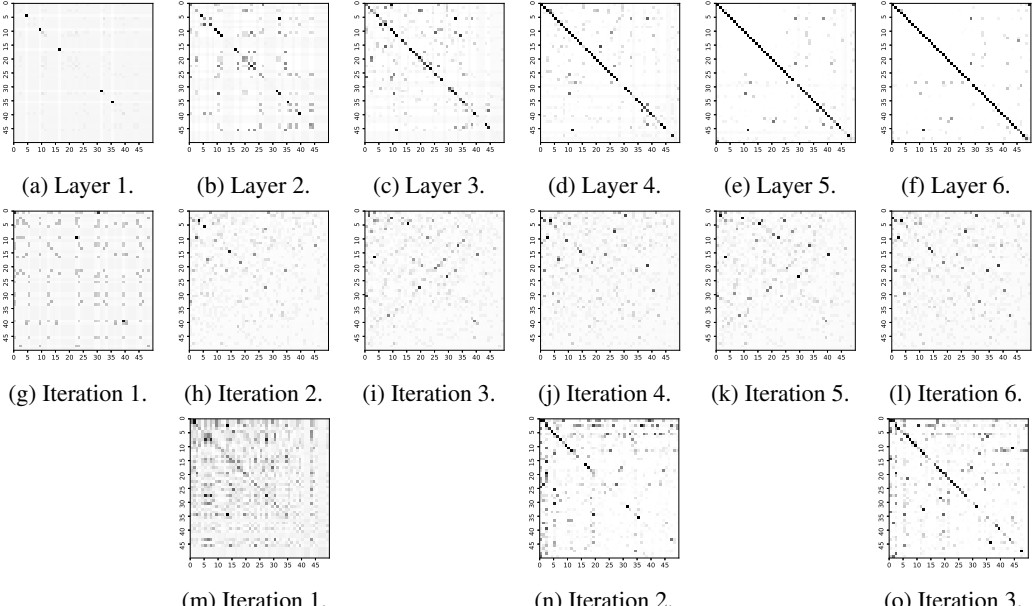

(a) Layer 1.  (b) Layer 2.  (c) Layer 3.  (d) Layer 4.  (e) Layer 5.  (f) Layer 6.

(g) Iteration 1.  (h) Iteration 2.  (i) Iteration 3.  (j) Iteration 4.  (k) Iteration 5.  (l) Iteration 6.

(m) Iteration 1.  (n) Iteration 2.  (o) Iteration 3.

Figure 11: The similarity/witness matrices of the matching process on a fixed pair of sparse correlated Erdős-Rényi graphs with $n = 50$, $p = 0.1$, $s = 0.8$ and $\theta = 0.1$. Darker points correspond to higher similarity (in $Y_l$) or a larger number of witnesses. Figure 11(a) — Figure 11(f) are the similarity matrix from each layer of SeedGNN. Figure 11(g) — Figure 11(l) are the witness matrix from each iteration of the iterative 1-hop algorithm. Figure 11(m) — Figure 11(o) are the witness matrix from each iteration of the iterative 2-hop algorithm.

In summary, from these two case studies, we conclude that our SeedGNN might be able to choose the appropriate features for different types of graphs according to the confidence level of new seeds. Further, we observe that the matching accuracy of SeedGNN is even higher than that of the 1-hop and 2-hop algorithms, the latter two of which have been theoretically proven to work well for dense graphs and sparse graphs, respectively (Mossel et al., 2019). Thus, this result suggests that SeedGNN may extract more valuable features, or learn more effective ways to synthesize witness information, than the theoretical algorithms.

## D  PERFORMANCE COMPARISON WITH GNN METHODS

In this section, we further compare the SeedGNN with several state-of-the-art deep graph matching networks, including semi-supervised learning methods (**PALE** (Man et al., 2016), **DeepLink** (Zhou et al., 2018), **dName** (Zhou et al., 2019), **CrossMNA** (Chu et al., 2019), **MGCN** (Chen et al., 2020) ) and supervised learning methods (**GMN** (Zanfir et al., 2018),**PCA-GM** (Wang et al., 2019),

**NGM** (Wang et al., 2021), **IPCA-GM** (Wang et al., 2020a), **CIE** (Yu et al., 2019), **GLMNet** (Jiang et al., 2022), **LCS** (Wang et al., 2020b), **DGMC** (Fey et al., 2020), **BB-GM** (Rolínek et al., 2020), **DGM** (Gao et al., 2021), **DLGM** (Yu et al., 2021c)). We conduct experiments on Willow Object dataset (Cho et al., 2013), which consists of 256 images in 5 categories. The training set contains all categories of images, with 20 images of each category. We test the trained models on the rest images. Following the experimental setups in (Fey et al., 2020), we construct graphs via the Delaunay triangulation of keypoints, and the input features of keypoints are given by the concatenated output of relu4_2 and relu5_1 of a pre-trained VGG16 (Simonyan & Zisserman, 2015). Note that the resulting graphs only have 10 nodes. For semi-supervised methods, we randomly choose 5 true pairs as seeds. For supervised methods, they do not need seeds. For SeedGNN, we still directly use the model trained in Section 5.1. We generate the seeds in two ways. The first way is to apply the Hungarian algorithm on the similarities of non-topological node features. The second way is to use the matching result of the GNN methods for seedless graph matching (we choose DGMC). Note that for both ways, our SeedGNN does not utilize any training graph in the Willow Object dataset. For the semi-supervised algorithms, we use the publicly available implementations from their respective papers to generate the corresponding matching results. The performance values of the existing supervised algorithms are directly retrieved from their respective papers.

Since there are lack of sufficient training data for semi-supervised methods (there are only 5 seeds for each pair of graphs), it is difficult for them to learn to match the seeds effectively. As a result, we observe in Table 3 that SeedGNN significantly outperforms the semi-supervised methods. In contrast, the supervised methods learn from a large number of graph pairs. Further, the two images to be matched are of the same category. Therefore, the input node feature generated are similar and informative enough for correlating keypoints. Thus, the supervised methods have performed relatively well, and SeedGNN using seeds generated by the non-topological node features does not achieve performance gain. However, we can use SeedGNN to refine the output of seedless graph matching algorithms. We observe that SeedGNN consistently improves the matching performance of DGMC and achieves the best performance.

Table 3: Comparison of matching accuracy (%) on Willow Object dataset. The best results are marked as bold. The performance values of the existing supervised algorithms are directly retrieved from their respective papers.

| | Method | face | mbike | car | duck | wbottle | Mean |
|---|---|---|---|---|---|---|---|
| Semi-Supervised | PALE (Man et al., 2016) | 85.4 | 52.9 | 55.1 | 56.4 | 68.1 | 60.1 |
| | DeepLink (Zhou et al., 2018) | 86.1 | 55.8 | 63.7 | 62.0 | 72.3 | 66.0 |
| | dName (Zhou et al., 2019) | 86.9 | 58.3 | 65.3 | 66.0 | 77.7 | 68.8 |
| | CrossMNA (Chu et al., 2019) | 85.6 | 60.1 | 61.4 | 65.8 | 74.2 | 68.0 |
| | MGCN (Chen et al., 2020) | 87.2 | 63.0 | 67.5 | 67.2 | 78.1 | 72.6 |
| Supervised | GMN (Zanfir et al., 2018) | 98.1 | 65.0 | 72.9 | 74.3 | 70.5 | 76.2 |
| | PCA-GM (Wang et al., 2019) | **100.0** | 76.7 | 84.0 | 93.5 | 96.9 | 90.2 |
| | NGM (Wang et al., 2021) | 99.2 | 82.1 | 84.1 | 77.4 | 93.5 | 87.2 |
| | IPCA-GM (Wang et al., 2020a) | **100.0** | 77.7 | 90.2 | 84.9 | 95.2 | 89.6 |
| | CIE (Yu et al., 2019) | **100.0** | 90.0 | 82.2 | 81.2 | 97.6 | 90.2 |
| | GLMNet (Jiang et al., 2022) | **100.0** | 89.7 | 93.6 | 85.4 | 93.4 | 92.4 |
| | LCS (Wang et al., 2020b) | **100.0** | 99.4 | 91.2 | 86.2 | 97.9 | 94.9 |
| | DGMC (Fey et al., 2020) | **100.0** | 92.1 | 90.3 | 89.0 | 97.1 | 93.7 |
| | BB-GM (Rolínek et al., 2020) | **100.0** | 98.9 | 95.7 | 93.1 | 99.1 | 97.4 |
| | DGM (Gao et al., 2021) | **100.0** | 98.8 | 98.0 | 92.8 | 99.0 | 97.7 |
| | DLGM (Yu et al., 2021c) | **100.0** | 99.3 | 96.5 | 93.7 | **99.3** | 97.8 |
| | SeedGNN (ours) | **100.0** | 98.9 | 98.0 | 93.1 | 98.7 | 97.7 |
| | DGMC+ SeedGNN | **100.0** | **99.6** | **100.0** | **99.7** | 99.1 | **99.5** |

