# OpenReview forum: "SeedGNN: Graph Neural Network for Supervised Seeded Graph Matching"
_ICLR.cc/2023/Conference — Submitted to ICLR 2023_

### Official Review · Reviewer_7PdA · 2022-10-20

**Confidence:** 5
**Correctness:** 3
**Technical Novelty And Significance:** 2
**Empirical Novelty And Significance:** 2
**Recommendation:** 5

**Clarity, Quality, Novelty And Reproducibility:**

In my view, the novelty of this work is unquestionable since the seeded graph matching problem is hardly considered in the literature. But the clarity and quality of the work are not so satisfactory due to the weakness I find. As for reproducibility, I can not make a judgment since the authors do not plan to open-source their code.


**Strength And Weaknesses:**

**Strength:**
1. This paper focuses on the seeded graph matching problem, which is hardly considered in the literature, especially for existing deep graph matching works. Indeed, initial seeds can make models better handle graphs with a large number of nodes. Therefore, I think it's necessary and interesting to consider the seeded graph matching problem.
2. The authors propose a new architecture called SeedGNN consisting of two main components namely the convolution module and the percolation module, which can better handle the seeds and generate more new seeds.

**Weakness:**
1. The authors claim, "In contrast to previous approaches that apply GNNs separately to each graph, a key departure of our SeedGNN architecture is to apply the GNN jointly over two graphs and to learn a pair-wise similarity for each pair of nodes directly." While in the field of deep graph matching, similar approaches already exist, such as the SOTA BBGM[1] and NGM[2]. These methods utilize the single affinity matrix constructed from the two input graphs as the input to their GNN. So, what is the main difference between the proposed SeedGNN and existing deep graph matching works?
2. As for the comparison of SeedGNN and existing deep graph matching works, I agree that existing deep graph matching works hardly consider utilizing the initial seeds, but I think it is too absolutely to claim that the "node-based GNN approach faces significant difficulty in learning how to use seed information". For example, we can set the initial seeds as matched pairs and let the methods learn the other nodes, or we can modify node/edge affinities in the affinity matrix based on the matching seeds, or we can remove the initial seeds from the input graph to generate a smaller graph and let the methods work on the new graph. Anyway, I think there are many straightforward approaches to allow existing deep graph matching methods to use the same seed information as the SeedGNN, and it is arbitrary to make such a judgment in this paper.
3. For the results in Table 1, I hope the authors may consider my suggestion above and make a more fair comparison. I think the current results in Table 1 look weird that the results of all supervised methods (including SOTA methods in graph matching) are zeros. Making the supervised methods use the same seed information will be better.
4. The authors claim the proposed SeedGNN can automatically choose the effective features many times in the paper, but the proof looks not sufficient enough to me. The authors only prove it in the experiments: "From theoretical graph matching results, we have learned that we need to use witnesses at different numbers of hops for matching sparse graphs (p = 0.01) and dense graphs (p = 0.2). Thus, these results suggest that our SeedGNN can adaptively choose the appropriate features to match different types of graphs." While I do not think it can demonstrate the claim. Moreover, I think this claim should be demonstrated in theory, since the good performance in the experiments may have multiple reasons instead of the reason for choosing the right features claimed by the authors.
5. For the masking part of the proposed SeedGNN, the authors claim "we leverage an approach called masking to remove the noisy information and retain the cleaner information."  However, I think this masking approach is very similar to one existing graph matching paper in ICLR 2020 (Eq 17 of [3]), which also uses Hungarian as a mask for the output. Could the authors please show the difference between their approach and the ICLR 2020 work?
6. The proposed SeedGNN can generate new seeds by matched node pairs in the matching process, but I wonder if it will be troublesome when the newly generated seed is wrong by mistake. For example, if the generated seeds at the beginning of the matching process are wrong, would it damage the entire matching process? Maybe an error correction or regret mechanism should be considered.

**References**

[1] Rolínek, Michal, et al. "Deep graph matching via blackbox differentiation of combinatorial solvers." European Conference on Computer Vision. Springer, Cham, 2020.

[2] Wang, Runzhong, Junchi Yan, and Xiaokang Yang. "Neural graph matching network: Learning Lawler's quadratic assignment problem with extension to hypergraph and multiple-graph matching." IEEE Transactions on Pattern Analysis and Machine Intelligence (2021).

[3] Yu, Tianshu, et al. "Learning deep graph matching with channel-independent embedding and hungarian attention." International conference on learning representations. 2019.


**Summary Of The Paper:**

In this paper, the authors focus on the seeded graph matching problem, which is a variant of the graph matching problem with several pre-matched pairs as initial seeds. In existing works, the seeds are hardly considered especially in supervised graph matching works. They propose to apply the GNN jointly over both the two input graphs for matching and try to learn the pair-wise similarities. Specifically, their proposed SeedGNN contains two modules: the convolution part for learning the witnesses to distinguish the true pairs and the fake pairs; the percolation part for filtering the prediction of the model and generating the new seeds by matched node pairs. Experiments are conducted on multiple datasets to verify the performance of the proposed SeedGNN with the comparison of various baselines including both semi-supervised learning and supervised learning.

**Summary Of The Review:**

This paper aims to solve the seeded graph matching problem which is hardly considered in the literature. However, except for this point, the other parts of this paper look not good enough to me. As I mentioned in the weakness part, the authors need to clarify the difference between several existing works to further demonstrate their contributions. Therefore, I consider this paper is not yet ready for publication in ICLR.

---

> ### Author Response · Authors · 2022-11-18
> **Response to Reviewer 7PdA, weakness 4-6**
>
> ## 4. Choosing approximate features
> Thank you for your comment! We agree that we cannot prove with certainty that our SeedGNN has chosen appropriate features, and thus our claim is too strong. The improved matching results seem to suggest that our SeedGNN has chosen better features, and our experiments in Appendix B.3 hope to provide more evidence by looking inside the layers. However, as in other multi-layer neural networks, it is difficult to know exactly what features have been learned by SeedGNN. We will tone down this statement in the revision as
>
> >"Thus, these results suggest that our SeedGNN chooses the appropriate features to match different types of graphs."
>
>
> ## 5. Masking with the Hungarian algorithm
> Thank you for the reference! We will cite this paper when we introduce masking in the revision. We also wish to point out that we use the Hungarian algorithm in a different way from that in [4]. [4] only uses the Hungarian algorithm in the loss function (Eq 14). The results of intermediate layers are still very noisy.
> We use the Hungarian algorithm in each layer to filter out the misleading information, and thus the final result would be better.
>
> >"[4] also uses the Hungarian algorithm for seedless graph matching, but they only clean up the matching result in their loss function. The results of intermediate layers are still very noisy. We use the Hungarian algorithm in each layer to filter out the misleading information, and thus the final result would be better."
>
> See Appendix A in the revision.
>
> [4] Yu, Tianshu, et al. "Learning deep graph matching with channel-independent embedding and hungarian attention." International conference on learning representations. 2019.
>
> ## 6. Incorrect Seeds
> Thank you for your comment! This is actually a crucial difference between our percolation module and the standard ways of percolation. Unlike previous percolation algorithms, we allow SeedGNN to correct errors by re-matching nodes at each layer. Note that in many percolation algorithms (e.g., [5]), once a new pair of seeds is identified, it will be used as the correct matching until the end. This approach can be problematic if an incorrect pair is identified as seeds, whose impact will be lasting for many iterations down the road. In contrast, since our SeedGNN rematches nodes at each layer, even if some of the newly-identified seeds in the previous layer are incorrect, we can potentially correct these errors in the next layer, as long as the fraction of incorrect seeds is small. In other words, our design of SeedGNN takes advantage of the power of partially-correct (i.e., noisy) seeds (as theoretically verified in [6]). In the revision, we will explain this crucial difference in Appendix A.
>
> [5] Rolínek, Michal, et al. "Deep graph matching via blackbox differentiation of combinatorial solvers." European Conference on Computer Vision. Springer, Cham, 2020.
>
> [6] Wang, Runzhong, Junchi Yan, and Xiaokang Yang. "Neural graph matching network: Learning Lawler's quadratic assignment problem with extension to hypergraph and multiple-graph matching." IEEE Transactions on Pattern Analysis and Machine Intelligence (2021).
>
> ## 7. Reproducibility
> Sorry for any confusion but we have provided our code in Supplementary Material. We will also open-source our code after our paper is accepted.

---

> ### Author Response · Authors · 2022-11-18
> **Response to Reviewer 7PdA, weakness 2-3**
>
> ## 2. Difficulties of node-based GNNs
>
> Thank you for your constructive comment! When we started this project, we actually began with node-based GNN, using similar ideas as what you suggested to add seed information. It is through the difficulty from our experience that we in the end decided to switch to pair-wise GNN. The discussion in Section 4.1 aims to convey the lesson that we learned from our experience. Specifically,
>
> 1. We believe that "adding node/edge affinity" is similar to the "cross-link" idea that we presented in Section 4.1. As we discussed there, the advantage of this encoding method is that it is easily generalizable to arbitrary test graphs. The difficulty for node-based GNN, however, is that this encoding only tells a node-based GNN that there is a seed in the neighborhood, but not the identity of the seed. In contrast, pair-wise GNN can utilize such encoding much more effectively.
>
> 2. We believe that "setting initial seeds as matched pairs" may be similar to the "one-hot encoding" or "random vector encoding" that we discussed in Section 4.1. The difficulty of this approach, however, is that the GNN model must change as the number of seeds in the test graph increases. As a result, it will be difficult for node-based GNN to generalize to test graphs with much larger number of seeds.
>
> 3. We have not experimented with the idea of "remove the initial seeds from the input graph to generate a smaller graph and let the methods work on the new graph". However, we have already found that supervised seedless matching algorithms will suffer poor performance when the non-topological node features are not informative (see Section 5.2). Thus, we feel that this approach of removing seeded nodes may not help answering the question of how to best use seed information.
>
> In summary, the intention of Section 4.1 is to convey to the readers why, from our own experience, it is not easy to design a node-based GNN that works well for seeded graph matching. However, we agree with the reviewer that our claim came across as being too general and absolute, as we obviously cannot enumerate all the possible ways to design node-based GNN with seeded information. In the revision, we will tone down this claim as
>
> >"Since most existing approaches for graph matching use node-based GNNs, we also started with node-based GNNs when we research the supervised approach to seeded-graph matching. However, from our experience, we found that node-based GNNs have significant difficulty effectively utilizing seed information."
>
> and focus instead on sharing the lesson that we learn from our own experience, which indeed motivates our direction of using pair-wise GNN. We thank you again for your comment.
>
> ## 3. Experimental comparison with supervised seedless algorithms
>
> We apologize for the confusion. The reason that we include results for supervised seedless matching algorithms here is to confirm that the non-topological node features in the SHREC'16 dataset are not informative. (They are merely 3D coordinates.) Thus, supervised seedless matching algorithms relying on non-topological node features will suffer poor performance. This explains why some semi-supervised algorithms for seeded graph matching (including CrossMNA and MGCN), which would have worked well when there are high-quality non-topological node features, also perform poorly for the SHREC'16 dataset.  Our original statement didn't make this intention clear, and hence may come across as being unfair. We will clarify our intention in Section 5.2 in the revision. Further, we agree that existing supervised models can also be supplemented with seed information. In the revision, we will add additional experimental results for the supervised models using random encoding to represent seeds in Section 5.2.

---

> ### Author Response · Authors · 2022-11-18
> **Response to Reviewer 7PdA, weakness 1**
>
> Thank you for carefully reading our paper and providing insightful feedback! Below, we address the specific comments/concerns raised. The changes in the paper are colored in red.
>
> ## 1. pair-wise GNN is not new
> We apologize for not discussing these references thoroughly in the original submission. We have now carefully revised several parts of the paper to compare with these two references. Although both of them also use the affinity matrix and [2] even uses a pair-wise architecture, we believe that our design of SeedGNN is still more effective and efficient:
>
> 1. Although [1] uses the affinity matrix, we believe that it is still a node-based GNN. The reason is that [1] does not use the affinity matrix in their GNN. Instead, they apply node-based GNNs on each graph to construct the affinity matrix. Then, they use dual ascent algorithms on the affinity matrix to solve the graph matching problem. Thus, their method still corresponds to a node-based GNN method and relies on high-quality node features. In contrast, we use a pair-wise GNN architecture that directly uses affinity information inside our GNN. In the revision, we have cited this paper and clarified the difference in Section 2 "Further Related Work".

---

> > ### Author Response · Authors · 2022-11-18
> > **cont.**
> >
> > 2. The architecture of [2] indeed shares considerable similarity with SeedGNN, and it also uses a pair-wise GNN and uses the affinity matrix as input. However, note that [2] focuses on seedless graph matching. Therefore, the NGM architecture in [2] was not designed for seeded graph matching. For example, they do not aim to exploit important features such as witnesses. Further, the NGM algorithm has not been evaluated for seeded graph matching either. In the revision, we have transferred the NGM approach to seeded graph matching by modifying the affinity matrix to encode seed information, and we have added experiments to compare our SeedGNN with the NGM algorithm in Section 5.2. Through our own experiments, we found that the NGM algorithm in [2] does not generalize well when the test graph is with much larger size and node-degree than the training graph. Specifically, we trained our SeedGNN and NGM with the same training set described in Section 5.1. Then, we test the models on graphs with larger size than the training graphs. The matching accuracy results on sparse correlated ER graphs with $n=500, p=0.01, s =0.8$ are shown below, as a function of the fraction of seeds $\theta$:
> > | $\theta$  |  0.0 | 0.02 | 0.04 | 0.06 | 0.08 | 0.1 | 0.12 | 0.14 | 0.16 | 0.18 | 0.2 |
> > |---|---|---|---|---|---|---|---|---|---|---|---|
> > |  SeedGNN  | 0.002 | 0.1512 | 0.474 | 0.8582 | 0.96 | 0.976 | 0.976 |  0.976 | 0.976 |0.976 | 0.976 |
> > |   NGM  |0.0035 | 0.0235 | 0.1401 | 0.4072 | 0.7628 | 0.9378 | 0.9604 | 0.9709 | 0.9718 | 0.9743 | 0.9747|
> >
> >    Note that both algorithms transition from low matching accuracy to nearly perfect matching accuracy around $\theta=0.08$ and $0.1$.
> >    In contrast, the matching accuracy results on dense correlated ER graphs with $n=500, p=0.2, s =0.8$ are shown below:
> >
> >    | $\theta$  |  0.0 | 0.005 | 0.01 | 0.015 | 0.02 | 0.025 | 0.03 | 0.035 | 0.04 | 0.045 | 0.05 |
> >    |---|---|---|---|---|---|---|---|---|---|---|---|
> >    |  SeedGNN  | 0.0012 | 0.0074 | 0.9136 | 1.0 | 1.0 | 1.0 | 1.0 | 1.0 | 1.0 | 1.0 | 1.0 |
> >    |   NGM  | 0.0013 | 0.005 | 0.012 | 0.018 | 0.021 | 0.032 | 0.058 | 0.43 | 0.93 | 1.0 | 1.0 |
> >
> >    We can see that SeedGNN transitions to nearly perfect matching accuracy at $\theta=0.015$, but NGM's transition requires $\theta=0.04$ (i.e., a much larger fraction of seeds). In summary, we have found that NGM performs close to our SeedGNN in larger sparse graphs ($p=0.01$) but performs poorly in larger dense graphs ($p=0.2$). One possible reason for this deterioration in the generalization power of NGM could be that, in the aggregation step, NGM normalizes each representation by the vertex degree of the association graph (which is roughly the square of the node degrees), but we do not. To see why this difference matters, note that according to known theoretical results on seeded graph matching, there exist algorithms that only need $\Omega(\log n)$ seeds to match all $n$ nodes [3]. However, if the graph sparsity $p$ is fixed, the node degree increases proportionally to $n$, and correspondingly the vertex degree of the association graph increases quadratically with $n$. As a result, when NGM divides the similarity of each node pair by the vertex degree, we expect that the resulting value (~$\frac{\log n}{n^2}$) will decrease close to zero as the graph size increases. Hence, it would be difficult for the sinkhorn step in NGM to distinguish true pairs from fake pairs in test graphs with larger size and node degrees than the training graphs. In contrast, since SeedGNN does not divide the similarity scores by the vertex degrees, the Hungarian algorithm step in our percolation module (which can distinguish any absolute difference) will then be able to distinguish the true pairs from the fake pairs. We have carefully revised our paper to explain this difference in Appendix A.1.
> >
> > [1] Rolínek, Michal, et al. "Deep graph matching via blackbox differentiation of combinatorial solvers." European Conference on Computer Vision. Springer, Cham, 2020.
> >
> > [2] Wang, Runzhong, Junchi Yan, and Xiaokang Yang. "Neural graph matching network: Learning Lawler's quadratic assignment problem with extension to hypergraph and multiple-graph matching." IEEE Transactions on Pattern Analysis and Machine Intelligence (2021).
> >
> > [3] Elchanan Mossel, Jiaming Xu, et al. Seeded graph matching via large neighborhood statistics. In Proceedings of the Thirtieth Annual ACM-SIAM Symposium on Discrete Algorithms, pp. 1005–1014. SIAM, 2019.

---

> > ### Comment · Reviewer_7PdA · 2022-11-18
> > **Thanks for your response, could you please update the PDF?**
> >
> > I thank the authors for the feedback, but it may take some time for me to fully review it. At this time, I am requesting the authors to highlight what has been changed in PDF (preferably by a different color), before the PDF revision deadline (there should be ~20hrs to go). It seems that many details have been updated, and I intend to evaluate the novelty, contributions, and correctness of this paper based on the new PDF.

---

> > > ### Author Response · Authors · 2022-11-18
> > > **The revised version of our paper has been updated**
> > >
> > > We have updated the revised paper. The changes in the paper are colored in red. Please let us know if you have any additional questions/suggestions. We thank you again for your constructive comments!

---

> > > > ### Comment · Reviewer_7PdA · 2022-11-22
> > > > **Post-rebuttal comments**
> > > >
> > > > Thank the authors again for the detailed responses, discussions, and new experimental results. Some of my concerns are relieved. After reading the revised version, I would like to share my concerns about the novelty of this paper.
> > > >
> > > > According to the introduction of this paper, this paper has the following contributions:
> > > > * A pair-wise GNN is developed for graph matching, which is different from other GNNs that work on single graphs, with the following designs for seeded graph matching:
> > > >   * The convolution module that learns to count “witnesses” at different hops
> > > >   * The percolation module that matches high-confidence node-pairs at one layer and propagates the matched node-pairs as new seeds to the subsequent layers, triggering a percolation process that matches a large number of node pairs
> > > >
> > > > However, as the authors will probably agree, the first main contribution is somewhat incremental given existing papers [1, 2]. The other two contributions do have some novelty under the context of seeded graph matching, but may not be strong enough to serve as the main contributions of an ICLR paper:
> > > >   * The convolution module propagates seed information just like other GNNs, and the difference is a tailored message passing function.
> > > >   * The way of adopting Hungarian in the percolation module can be found exactly in [4] under a very similar motivation (suppress matches that do not exist in the final matching result), though I agree that the application is somewhat different.
> > > >
> > > > Considering all the aforementioned aspects above, I intend to raise my score to 5 because the authors address my concerns, yet I still lean towards reject because the novelty of the current version seems still limited.

---

> > > > > ### Author Response · Authors · 2022-11-23
> > > > > **Reponse to post-rebuttal comments**
> > > > >
> > > > >
> > > > > We thank the reviewer for the further comments. If we may, we wish to follow up with two reasons why we respectfully disagree with the comment that our contributions are limited.
> > > > >
> > > > > * For the first concern, we believe that our successful design of pair-wise GNN for seeded graph matching *is* a significant contribution by itself. Specifically, [1] does not use a pair-wise GNN as we have discussed in Section 2. [2] uses a pair-wise GNN but only for seedless graph matching problems. Hence, it does not correctly account for important features such as witness, which is unique for seeded matching. Our new experimental results in Section 5.2 already show that, when the NGM architecture of [2] does not account for the important role of witness (especially when the number of seeds is low), it will lead to poor generalization capability for seeded graph matching. Thus, to the best of our knowledge, our work is the first in the literature that makes pair-wise GNN work for seeded graph matching. Part of the reason for the superior performance is because we utilize insights from prior theoretical results for seeded graph matching, which has not been done for GNN before. Therefore, we believe that our first contribution (of using pair-wise GNN for seeded graph matching) is significant.
> > > > >
> > > > > * About the use of the Hungarian algorithm in [4], we also respectfully disagree that "the way of adopting Hungarian... can be found exactly in [4]". As we explained earlier, the way that we use the Hungarian algorithm is very different from [4]:
> > > > > > "[4] also uses the Hungarian algorithm for seedless graph matching, but they only clean up the matching result in their loss function. The results of intermediate layers are still very noisy. We use the Hungarian algorithm in each layer to filter out the misleading information, and thus the final result would be better."
> > > > >
> > > > >   While the Hungarian algorithm has been a common building block for linear assignment problems, this way of using the Hungarian algorithm at the intermediate layers of GNN is new. It is important because the Hungarian algorithm at the intermediate layers can produce cleaner information to enhance/enrich the input data for the next layer. The Hungarian algorithm in the loss function does not have this capability.
> > > > >
> > > > > Please let us know if you have any additional questions/suggestions. We thank you again for your comments!

---

### Official Review · Reviewer_g881 · 2022-10-24

**Confidence:** 3
**Correctness:** 4
**Technical Novelty And Significance:** 3
**Empirical Novelty And Significance:** 2
**Recommendation:** 6

**Clarity, Quality, Novelty And Reproducibility:**

The paper is generally well-written and the ideas are clearly explained. The originality of the proposed method is that it integrates several graph matching techniques into a GNN-based differentiable model.

**Strength And Weaknesses:**

Strength:
1. The seeded graph matching problem is important, which has many applications.
2. The end-to-end learning framework is novel and the design is sophisticated but clearly explained.
3. The empirical results seems promising.
4. Comprehensive auxiliary experiments  are also provided, which are informative.

Weaknesses:
1. The ideas behind each building block are not new, which have been used in previous graph matching algorithms.
2. The main drawback of the framework is the high computational complexity. The complexity of the proposed model is essentially the same as a GNN model on a graph with $n_1n_2$ nodes and $m_1m_2$ edges. Thus the time and space complexity is $m_1m_2$ and $n_1n_2$  respectively for each layer. This is huge for moderate-size graphs.  I think the memory usage is the reason why the model is trained on CPU instead of on GPU.

**Summary Of The Paper:**

This paper studies the seeded graph matching problem, in which the learning algorithm is given a set of ground-truth seed pairs and the task is to match the rest of the nodes. The contribution of the paper is an end-to-end learning based on GNN. The model consists of two modules, namely convolution and percolation modules. The proposed method achieve competitive performance on real and synthetic datasets.

**Summary Of The Review:**

This paper proposes an end-to-end GNN model for seeded graph matching. The design of the model is interesting and achieve promising empirical results. However, the high time and space complexity of the model make it not suitable for matching large-scale graphs.

---

> ### Author Response · Authors · 2022-11-18
> **Response to Reviewer g881, weakness 2**
>
> ## 2. Complexity
>
> We acknowledge that the time complexity of our method is higher than the best-known time complexity of GNN-based algorithms for graph matching. However, we believe that the increase in complexity is moderate, for the following reasons:
>
> 1. Although we also mentioned $O(m_1m_2)$ complexity in our original manuscript, we have found that it is over-estimated. In the revision, we have reduced the time complexity of our method to $O(n_1n_2d_{\text{mean}})$, where $d_{\text{mean}}$ is the mean of node degrees of the matching graphs. Specifically, in Eq (2) of our paper, we rewrite the aggregation step in our SeedGNN as:
> >"$H_l= \mathsf{unvec}((\mathbf{A}_1\otimes \mathbf{A}_2)s_l)=\mathbf{A}_1 S_l\mathbf{A}_2$"
>
>     The time complexity of right-hand-side we reported is $O(n_1n_2^2)$ in our original manuscript. In the revision, we have found that its time complexity can be further reduced to $O(n_1n_2d_{\text{mean}})$. Specifically, $\mathbf{A}_1$ and $\mathbf{A}_2$ have $O(n_1d{\text{mean}})$  and $O(n_2d{\text{mean}})$ non-zero entries, respectively. Thus, with sparse matrix multiplication, the time complexity of the right-hand-side is only $O(n_1n_2d{\text{mean}})$. Therefore, our time complexity is lower than both $O(n_1n_2^2)$ and $O(m_1m_2)$, as reported in our original paper. See the discussion around Eq (2) in Section 4.2 in the revision.
>
>
> 2. The complexity of our method is only moderately higher than the best-known complexity. Specifically, to the best of our knowledge, the best-known time complexity is $O(n_1n_2)$ [3]. Thus, our new time complexity above is only higher by a factor $d_{\text{mean}}$. Further, in many real-world graphs, $d_{\text{mean}}$ is small [4]. Thus, the time complexity of our method can be comparable to the best-known one. Moreover, in our original paper, our numerical result in Section 5.2 has demonstrated that the run time of our SeedGNN is comparable to the best-known GNN-based algorithms.
>
> 3. The reason that we do not use GPU for training is not because we cannot train on large graphs, but instead because we do not need to. By training on small graphs, our supervised approach already learns enough generalizable knowledge to match large graphs. Please see the discussions in Section 5.2. Hence, we do not experience complexity issues during training.
>
> [3] Matthias Fey, Jan E. Lenssen, Christopher Morris, Jonathan Masci, and Nils M. Kriege. Deep graph
> matching consensus. In International Conference on Learning Representations, 2020.
>
> [4] Albert-László Barabási. 2016. Network Science. Cambridge University Press, Cambridge.

---

> ### Author Response · Authors · 2022-11-18
> **Response to Reviewer g881, Weakness 1**
>
> We thank the reviewer for the comments. Below, we address the specific comments/concerns raised. The changes in the paper are colored in red.
>
> ## 1. Ideas are not new
> Although both "witness" and "percolation" are inspired by previous graph matching algorithms, there are a number of crucial and non-trivial differences in our design of SeedGNN.
>
> 1. Our convolution module generalizes the concept of "number of witnesses" from integer values to any real values. Specifically, in most prior work based on the notion of witness, they count an (integer) number of whiteness in an $l$-hop neighborhood. In contrast, in the second layer and above, the input to our convolution module already contains new seed information (from the percolation module) that carries real values (which correspond to the confidence of the new seeds). As a result, the "counting" of witnesses in (Eq 1) will also produce richer real values. This generalization greatly enhances the ability of SeedGNN to measure and synthesize similarity with different levels of confidence.
>
> 2. Unlike previous percolation algorithms, we allow SeedGNN to correct errors by re-matching nodes at each layer. Note that in many percolation algorithms (e.g., [1]), once a new pair of seeds is identified, it will be used as the correct matching until the end. This approach can be problematic if an incorrect pair is identified as seeds, whose impact will be lasting for many iterations down the road. In contrast, since our SeedGNN rematches nodes at each layer, even if some of the newly-identified seeds in the previous layer are incorrect, we can potentially correct these errors in the next layer, as long as the fraction of incorrect seeds is small. In other words, our design of SeedGNN takes advantage of the power of partially-correct (i.e., noisy) seeds (as theoretically verified in [2]). Second, as we discussed in Introduction and Section 4.3, most previous GNN-based methods for graph matching only use the soft-correspondence $Y_l$ as new seed information for the next layer. However, the soft-correspondence is very noisy, and directly utilizing such information would lead to even more matching errors. Our design of the percolation module carefully filters out such noisy information (by the masking step).
>
> [1] Lyudmila Yartseva, Matthias Grossglauser, et al. On the performance of percolation graph matching.
> In Proceedings of the first ACM conference on Online social networks, pp. 119–130. ACM, 2013.
>
> [2] Liren Yu, Jiaming Xu, and Xiaojun Lin. Graph Matching with Partially-Correct Seeds. J. Mach. Learn. Res., 2021, 22: 280:1-280:54.

---

### Official Review · Reviewer_fUFE · 2022-10-30

**Confidence:** 4
**Correctness:** 3
**Technical Novelty And Significance:** 2
**Empirical Novelty And Significance:** 2
**Recommendation:** 5

**Clarity, Quality, Novelty And Reproducibility:**

Overall clarity is fine except the explanation of witnesses. Novelty may not be adequate.

**Strength And Weaknesses:**

Strong points:

S1. The proposed method is able to learn an inductive model for seeded graph matching, instead of traditionally transudative approaches.

S2. Experiments are conducted on both synthetic and real-world datasets

Weaknesses:

W1. Inductive semi-supervised learning has been proposed in other tasks before, e.g. node classification [1]. Despite the task differences, the model design might still be relevant and some ideas can be borrowed. At the minimum, a discussion on the model design should be included to highlight why previous inductive semi-supervised approaches cannot be adopted here.

[1] Meta-Inductive Node Classification across Graphs  https://arxiv.org/abs/2105.06725

W2. The idea of using easily matched pairs as new seeds in the percolation layer is quite straightforward and used in other seeded problems, where a few initial seeds can bootstrap the process and high-confidence instances can be added as new seeds later.

W3. The "witness" is an important concept. While i get the general idea at the high level, it is not clearly defined: what exactly are witnesses? Is there a precise definition?

W4. In table 1, the supervised seedless models have very low accuracy (0-0.1%). But is such comparison meaningful? After all, the supervised methods are not designed for such seeded setting with very small supervision. Some other models, eg, self-supervised/contrastive models, may better handle small labeled data.

**Summary Of The Paper:**

This paper proposes a GNN for seeded graph matching. In the non-seeded version, the goal is to align/match the nodes between two graphs (eg. in social networks/ego networks, the same users appear across multiple networks.), while in the seeded version, there are a small number of pre-existing matching given. Most previous GNNs for seeded graph matching adopt a transductive semi-supervised setting, which  cannot learn transferable prior that is generalizable to unseen graphs. This paper proposes SeedGNN, which operates on a pair-level. There is first a convolution module to capture witnesses of seeds across different hops, and a percolation module to filter high-confidence pairs and use them as new seeds. SeedGNN is evaluated on both synthetic and real graphs.




**Summary Of The Review:**

Overall, I think the weaknesses overweigh the strengths. The weaknesses should be clarified/corrected.

---

> ### Author Response · Authors · 2022-11-18
> **Response to Reviewer fUFE, w2-4**
>
> ## 2. Percolation is straightforward
>
> We respectfully disagree that our use of the percolation idea is straightforward. Although we use the name "percolation module" since it is inspired by existing graph matching algorithms, there are several crucial and non-trivial differences in the design of this module.
>
> 1. Unlike previous percolation algorithms, we allow SeedGNN to correct errors by re-matching nodes at each layer. Note that in many percolation algorithms (e.g., [4]), once a new pair of seeds is identified, it will be used as the correct matching until the end. This approach can be problematic if an incorrect pair is identified as seeds, whose impact will be lasting for many iterations down the road. In contrast, since our SeedGNN rematches nodes at each layer, even if some of the newly-identified seeds in the previous layer are incorrect, we can potentially correct these errors in the next layer, as long as the fraction of incorrect seeds is small. In other words, our design of SeedGNN takes advantage of the power of partially correct (i.e., noisy) seeds (as theoretically verified in [5]). In the revision, we will explain this crucial difference in Appendix A.
>
> 2. As we discussed in Introduction and Section 4.3, most previous GNN-based methods for graph matching only use the soft-correspondence $Y_l$ as new seed information for the next layer. However, the soft-correspondence is very noisy, and directly utilizing such information would lead to even more matching errors. Our design of the percolation module carefully filters out such noisy information (by the masking step).
>
> [4] Lyudmila Yartseva, Matthias Grossglauser, et al. On the performance of percolation graph matching.
> In Proceedings of the first ACM conference on Online social networks, pp. 119–130. ACM, 2013.
>
> [5] Liren Yu, Jiaming Xu, and Xiaojun Lin. Graph Matching with Partially-Correct Seeds. J. Mach. Learn. Res., 2021, 22: 280:1-280:54.
>
> ## 3. The concept of witness
> We apologize for the confusion. We will add a more precise definition of witness in Section 4.2 in the revision.
>
> >" Given any graph $\mathcal{G}$ and two nodes $u,v$ in $\mathcal{G}$, we denote the length of the shortest path from $u$ to $v$ in $\mathcal{G}$ by $\text{dist}_{\mathcal{G}}(u,v)$. Then, for each node pair $(u,v)$ with $u$ in $\mathcal{G}_1$ and $v$ in $\mathcal{G}_1$, the seed $(w,\pi(w))$ becomes a $l$-hop witness for  $(u,v)$ if  $\text{dist}{\mathcal{G}_1}(u,w)=l$ and $\text{dist}{\mathcal{G}_2}(v,\pi(w))=l$."
>
>
> ## 4. Numerical comparison with seedless matching algorithms
> We apologize for the confusion. The reason that we include results for supervised seedless matching algorithms here is to confirm that the non-topological node features in the SHREC'16 dataset are not informative. (They are merely 3D coordinates.) Thus, supervised seedless matching algorithms relying on non-topological node features will suffer poor performance. This explains why some semi-supervised algorithms for seeded graph matching (including CrossMNA and MGCN), which would have worked well when there are high-quality non-topological node features, also perform poorly for the SHREC'16 dataset.  Our original statement didn't make this intention clear, and hence may come across as being unfair. We will clarify our intention in Section 5.2 in the revision. Further, we agree that existing supervised models can also be supplemented with seed information. In the revision, we will add additional experimental results for the supervised models using random encoding to represent seeds in Section 5.2.

---

> ### Author Response · Authors · 2022-11-18
> **Response to Reviewer fUFE, w1**
>
> We thank the reviewer for the comments. Below, we will address the detailed concerns raised by the reviewer. The changes in the paper are colored in red.
>
> ## 1. Inductive semi-supervised approaches
>
> Thank you for the reference! It would be an interesting future direction to explore how our proposed SeedGNN can be enhanced with the meta-learning approach in [1]. Indeed, our goal of using supervised learning for seeded graph matching shares some similarity with the work in [1], which also aims to both perform inductive learning (i.e., learn transferrable knowledge from training graphs) and utilize a small amount of labeled data on the test graph. However, [1] focuses on a node classification problem, which is quite different from seeded graph matching. In particular, [1] uses node-based GNNs, which (as we discussed in Section 4.1) have more difficulty in effectively utilizing seed information than our proposed pair-wise GNN. Further, in order to transfer knowledge from the trained GNN to test graphs, [1] scales all GNN weights by a common factor. It is unclear how this scaling will effectively transfer knowledge for seeded graph matching, e.g., how to best use different hops of witnesses. In contrast, our design of SeedGNN exploits the inherent structure of the seeded graph matching problem, and can be shown to generalize well to unseen graphs of sizes and types very different from the training set. For future work, it would be of interest to explore whether our SeedGNN can be further improved with a meta-learning component. In the revision, we discuss the inductive semi-supervised GNN approaches in Appendix A "Additional Related Work".
>
> >"Our goal of using supervised learning for seeded graph matching shares some similarity with the work in [1], which also aims to both perform inductive learning (i.e., learn transferrable knowledge from training graphs) and utilize a small amount of labeled data on the test graph. However, [1] focuses on a node classification problem, which is quite different from seeded graph matching. In particular, [1] uses node-based GNNs, which (as we discussed in Section 4.1) have more difficulty in effectively utilizing seed information than our proposed pair-wise GNN. Further, in order to transfer knowledge from the trained GNN to test graphs, [1] scales all GNN weights by a common factor. It is unclear how this scaling will effectively transfer knowledge for seeded graph matching, e.g., how to best use different hops of witnesses. In contrast, our design of SeedGNN exploits the inherent structure of the seeded graph matching problem, and can be shown to generalize well to unseen graphs of sizes and types very different from the training set. For future work, it would be of interest to explore whether our SeedGNN can be further improved with a meta-learning component [2]."
>
> [1] Wen Z, Fang Y, Liu Z. Meta-inductive node classification across graphs. Proceedings of the 44th International ACM SIGIR Conference on Research and Development in Information Retrieval. 2021: 1219-1228.
>
> [2] Adam Santoro, Sergey Bartunov, Matthew Botvinick, Daan Wierstra,and Timothy Lillicrap. Meta-learning with memory-augmented neural networks. In ICML, 2016, 1842–1850.

---

### Decision · Program_Chairs · 2023-01-20

**Decision:**

Reject

**Justification For Why Not Higher Score:**

* Two reviewers note that inductive semi-supervised learning has been proposed in other tasks before. Despite the task differences, the model design might still be relevant. Some ideas can be borrowed.
* The reviewers also agree that ideas behind each building block are not new, and have been used in previous graph-matching algorithms.
* Finally, the main drawback of the framework is the high computational complexity. The complexity of the proposed model is essentially the same as a GNN model on a graph with nodes and edges. Thus the time and space complexity increase considerably in each GNN layer. This is huge for moderate-size graphs and has been acknowledged by authors in the rebuttal.

**Justification For Why Not Lower Score:**

N/A

**Metareview: Summary, Strengths And Weaknesses:**

This paper describes a GNN for seeded graph matching. Given a small number of pre-existing matching given, the goal is to align/match the nodes between two graphs. Previous GNNs for seeded graph matching use a transductive semi-supervised setting. In contrast, SeedGNN operates on a pair level using a convolution module to capture seeds across different hops and a percolation module to filter high-confidence pairs and use them as new seeds.

**Strengths:**
* SeedGNN is evaluated on both synthetic and real graphs.
* SeedGNN is inductive and can learn transferable prior that is generalizable to unseen graphs.

**Weaknesses:**
* Two reviewers note that inductive semi-supervised learning has been proposed in other tasks before. Despite the task differences, the model design might still be relevant. Some ideas can be borrowed.
* The reviewers also agree that ideas behind each building block are not new, and have been used in previous graph-matching algorithms.
* Finally, the main drawback of the framework is the high computational complexity. The complexity of the proposed model is essentially the same as a GNN model on a graph with nodes and edges. Thus the time and space complexity increase considerably in each GNN layer. This is huge for moderate-size graphs and has been acknowledged by authors in the rebuttal.